# Placental growth factor exerts a dual function for cardiomyogenesis and vasculogenesis during heart development

Nevin Witman[1,5], Chikai Zhou[1,5], Timm Häneke[1], Yao Xiao[1], Xiaoting Huang[1], Eduarde Rohner [1], Jesper Sohlmér [1], Niels Grote Beverborg [1,2], Miia L. Lehtinen[1,3], Kenneth R. Chien [1,6] ✉ & Makoto Sahara [1,4,6] ✉

Cardiogenic growth factors play important roles in heart development. Placental growth factor (PLGF) has previously been reported to have angiogenic effects; however, its potential role in cardiogenesis has not yet been determined. We analyze single-cell RNA-sequencing data derived from human and primate embryonic hearts and find PLGF shows a biphasic expression pattern, as it is expressed specifically on ISL1+ second heart field progenitors at an earlier stage and on vascular smooth muscle cells (SMCs) and endothelial cells (ECs) at later stages. Using chemically modified mRNAs (modRNAs), we generate a panel of cardiogenic growth factors and test their effects on enhancing cardiomyocyte (CM) and EC induction during different stages of human embryonic stem cell (hESC) differentiations. We discover that only the application of PLGF modRNA at early time points of hESC-CM differentiation can increase both CM and EC production. Conversely, genetic deletion of *PLGF* reduces generation of CMs, SMCs and ECs in vitro. We also confirm in vivo beneficial effects of PLGF modRNA for development of human heart progenitor-derived cardiac muscle grafts on murine kidney capsules. Further, we identify the previously unrecognized PLGF-related transcriptional networks driven by EOMES and SOX17. These results shed light on the dual cardiomyogenic and vasculogenic effects of PLGF during heart development.

Heart disease is a leading cause of mortality worldwide with insufficient therapeutic options and poor prognosis. This is mainly derived from the fact that the mammalian heart has minimal regenerative capabilities among tissues and organs, most of which are lost after birth[1,2]. Thus, developing a new therapeutic strategy for heart regeneration is a major goal but the most critical challenge in modern cardiac biology and medicine. Although advances in stem cell biology including human pluripotent stem cell (hPSC) technology-based cell therapies hold great promise in regenerative medicine, therapeutic effects of the cell therapies for deficit hearts remain elusive, partly due to low functional engraftment of transplanted cells into the host myocardium. Therefore, these attempts are still in their infancy and far from clinical application[3,4]. On the other hand, recent sophisticated single-cell omics studies of in vitro human embryonic stem cell (hESC)-derived cardiac cells and in vivo human embryonic heart cells have uncovered cellular heterogeneity and molecular signatures, and identified previously

[1]Department of Cell and Molecular Biology, Karolinska Institutet, A6 Biomedicum, SE-171 77 Stockholm, Sweden. [2]Department of Cardiology, University Medical Center Groningen, University of Groningen, Groningen, The Netherlands. [3]Department of Cardiac Surgery, Heart and Lung Center, Helsinki University Hospital and University of Helsinki, Helsinki, Finland. [4]Department of Surgery, Yale University School of Medicine, 333 Cedar Street, New Haven, CN 06510, USA. [5]These authors contributed equally: Nevin Witman, Chikai Zhou. [6]These authors jointly supervised this work: Kenneth R. Chien, Makoto Sahara. ✉e-mail: kenneth.chien@ki.se; makoto.sahara@ki.se

unrecognized heart progenitors and cardiogenic molecules that would play certain roles in developing hearts[5–7]. In fact, a wide variety of paracrine mediators, such as growth factors, cytokines/chemokines, and secreted proteins, play important roles in heart development through regulating the differentiation and proliferation of cardiac progenitors and their progeny cells[8,9]. Some of these specific factors appear to have a direct or indirect function in inducing the production of cardiomyocyte (CM) and/or vascular cells within developing hearts, and thereby such factors may possess the capabilities to promote cardiac regeneration and repair in the diseased settings. Placental growth factor (PLGF), a member of the vascular endothelial growth factor (VEGF) family, was initially detected in the placenta and also found to a lesser extent across other tissues such as skeletal muscles and the heart[10]. PLGF binds to both soluble and non-soluble forms of FMS-related tyrosine kinase-1 (FLT1) (also termed vascular endothelial growth factor receptor-1 [VEGFR1]), and has previously been reported to have angiogenic effects in healthy and damaged hearts[11,12]. However, its potential role in cardiogenesis has not yet been determined.

Chemically modified mRNA (modRNA) is a synthetic and biocompatible molecule for efficient, dose-dependent, and transient protein expression in vitro and in vivo with low innate immunogenicity[13,14]. In the field of cardiovascular medicine, the VEGF-A modRNA was previously shown to have therapeutic effects in ischemia-related cardiac injury, through regenerative angiogenesis in animal and human models[15,16]. The modRNA-specific pharmacokinetics, which induces pulse-like but non-sustained protein expression, may offer an advantage in the aspects of inducing more potent biological effects and/or reduced side effects for the treatment of cardiovascular diseases[17].

Here, using an approach that employs single-cell RNA-sequencing (RNA-seq) datasets derived from human and primate embryonic hearts[18], we show atlases of cardiogenic growth factor expression profiles within heart cell lineages (e.g., second and first heart field [SHF/FHF] progenitors, CMs, pacemaker cells, smooth muscle cells [SMCs], endothelial cells [ECs] and cardiac fibroblasts) in developing hearts. Of interest, we find that PLGF exhibits a biphasic expression pattern during cardiogenesis, as it is expressed specifically on ISL1[+] SHF progenitors at the earlier embryonic stage and then on vascular cells such as SMCs and ECs at the later embryonic stage. We generate modRNAs of the 24 representative cardiogenic growth factors and test their efficacies to enhance CM and EC induction in in vitro hESC differentiation assays, which show that only PLGF modRNA increases both CM and EC production in vitro. Conversely, CRISPR-Cas9-mediated genetic deletion of *PLGF* significantly reduces the generation of vascular cells such as ECs and SMCs and also affects CM induction and its maturation in in vitro hESC differentiation. We further confirm the in vivo cardiomyogenic and vasculogenic effects of PLGF for the development of human heart progenitor-derived cardiac muscle grafts on murine kidney capsules[19]. These effects of PLGF are supported by the RNA-seq data and gene set enrichment analysis using in vitro hESC-derived cardiac cells. Further, through chromatin immunoprecipitation (ChIP) assays, we identify the previously unrecognized interactions between a cardiac precursor-specific transcription factor (TCF) EOMES[20] and its potential target PLGF at the earlier developing stage, as well as between an EC lineage-specific TCF SOX17[21] and its potential target PLGF at the later developing stage. Overall, these results shed light on the sequential cardiomyogenic and vasculogenic effects of PLGF in cardiogenesis and suggest a therapeutic potential for PLGF modRNA in heart disease.

## Results

### Embryonic heart single-cell RNA-seq analyses define atlases of cardiogenic growth factors

To clarify the roles and associations of each of the growth factors in heart development, we first utilized a single-cell RNA-seq dataset, previously obtained from human embryonic heart samples (4.5 to

10 weeks of fetal ages)[6,18]. A total of the 458 individual cardiac cells derived from micro-dissected heart regions, i.e., outflow tract (OFT), atria, and ventricles were segregated into 10 clusters including a cono-ventricular region-specific heart progenitor (CVP; cluster #1) that appeared predominantly in OFT at earlier stages (4.5 to 5.5 weeks of fetal age), by a dimensionally reduction method such as t-distributed stochastic neighbor embedding (tSNE) based on the profiles of differentially expressed genes (Supplementary Fig. 1a, b). Expression of the early cardiogenic and SHF progenitor-specific genes, such as *ISL1*[22], *BMP4*[23], *MEIS2*[24], *PDGFRA*[25], and *LGR5*[6] were enriched in the cells of the CVP cluster (Supplementary Fig. 1c, d), suggesting that the CVPs would represent more immature heart progenitors through the stages. To assess the significance of the co-expression of genes in cells, we performed Guilt-by-Association and correlation analysis[26]. Among representative 52 growth factors (Supplementary Data 1), we identified the top 12 growth factors that were highly co-expressed in the typical SHF/OFT progenitor *ISL1*[+] cells, indicating the specific roles of these factors in the human early heart progenitors (Supplementary Fig. 1e).

To gain further insights regarding the roles and associations of growth factors in cardiogenesis, we next analyzed single-cell RNA-seq datasets of the obtained non-human primate embryonic hearts (Methods). Similarly, the harvested hearts were micro-dissected as a whole heart (4 weeks of fetal age) or into OFT, atria, and right and left ventricle (RV and LV) (7 weeks of fetal age), and single cardiac cells dissociated from each of the compartments were obtained using a fluorescence-activated cell sorter. The gating strategies for sorting the live cells are shown in Supplementary Fig. 2a. Then, through single-cell RNA-seq analysis, a total of 1786 individual cardiac cells derived from the primate embryonic hearts were divided into 13 clusters, including the SHF/OFT progenitors (cluster #5), by tSNE using the Seurat program[27] (Fig. 1a, b). Differential gene expression analysis revealed that the pan-cardiac and FHF-related markers, such as *NKX2-5*, *HAND1*, *IRX4*, and *TNNT2*[28,29] were enriched in clusters #2, #9, and #10, which were considered as late CMs, FHF progenitors, and proliferative CMs, respectively, whereas the SHF/OFT progenitor markers *ISL1*, *FGF10*, *NKX2-6*, and *HOXA1* were enriched in a cluster #5 (Fig. 1c, d). The EC markers such as *PECAM1* and *CDH5* were enriched in both clusters #8 and #11, while an endocardium marker *NPR3* was enriched only in a cluster #8, indicating that clusters #8 and #11 were considered as endocardium and ECs, respectively (Supplementary Fig. 3a, b). The neural crest cell (NCC) and SMC markers such as *NGFR* and *MYH11* were enriched in clusters #6 and #13, which were considered as NCCs and SMCs, respectively (Supplementary Fig. 3b). Expression patterns of genes specific to other clusters are also shown in Supplementary Fig. 3b.

Next, using Guilt-by-Association and correlation analysis for single-cell RNA-seq data of primate embryonic hearts, we examined the co-expression of each of growth factors with each of the typical markers for multiple heart cell types (i.e., SHF, FHF, CM, pacemaker cell, EC, SMC, and cardiac fibroblasts) and identified the growth factors specific to each of the cell types (Fig. 1e). Finally, we merged both differential gene expression analysis in the clusters defined by tSNE on the Seurat program (Fig. 1a, b) and Guilt-by-Association and correlation analysis (Fig. 1e), which established the atlas of growth factor expression profiles of multiple heart cell types in developing hearts (Fig. 1f). For example, expression of bone morphogenetic protein 5 (BMP5) and epidermal growth factor-like protein 7 (EGFL7) were strongly distributed to FHF/CM and ECs/endocardial cells, respectively, highlighting the functional roles of these factors in multiple heart cell types (Fig. 1f; Supplementary Fig. 3c, d).

### Biphasic expression patterns of placental growth factor in SHF progenitors and vascular cells

Based on the results in single-cell RNA-seq analyses of human and primate embryonic hearts (Fig. 1; Supplementary Figs. 1 and 3), we selected the 24 growth factors that appear to promote cardiomyogenesis and/or

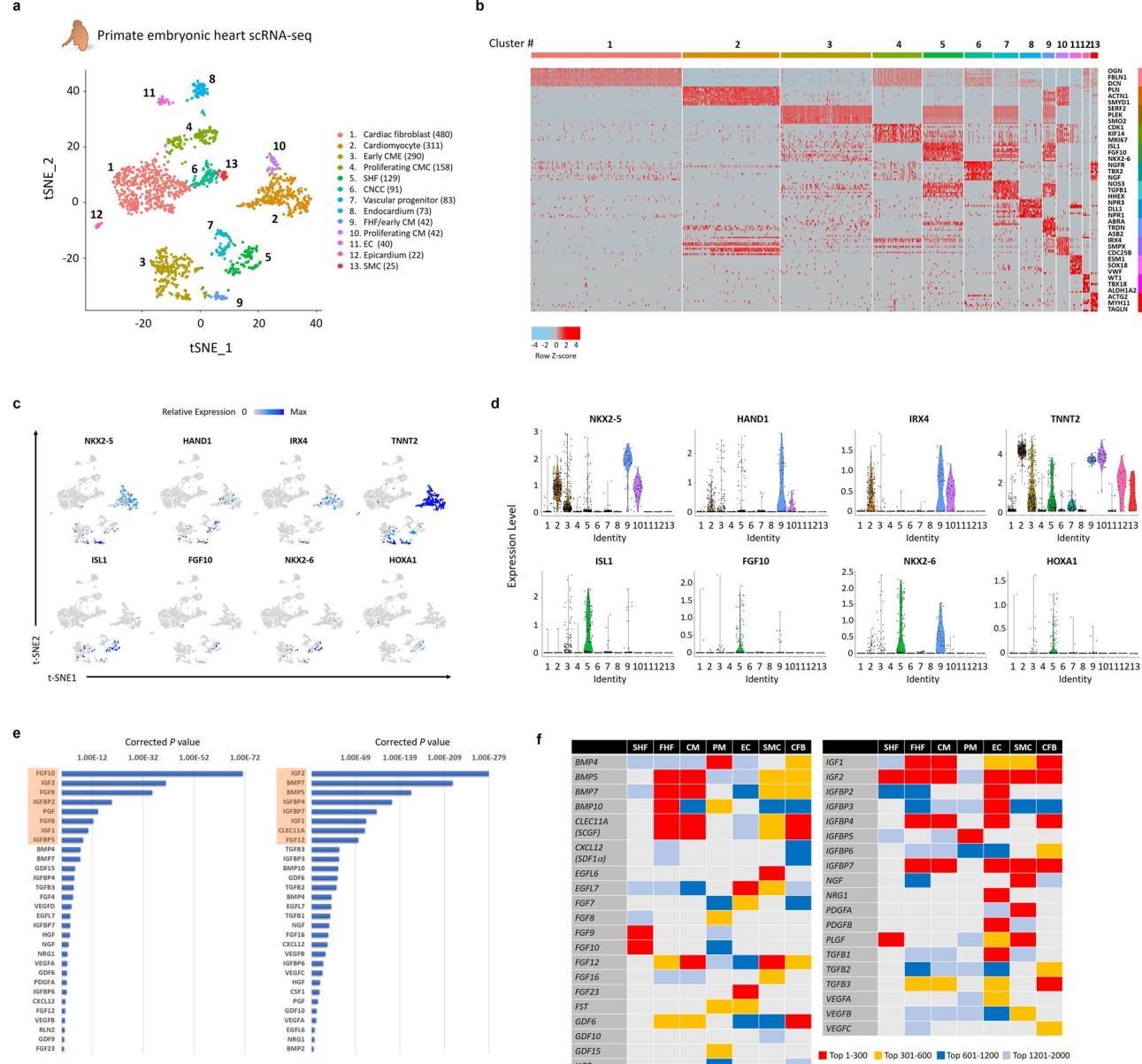

**Fig. 1 | Single-cell RNA-seq analysis of primate embryonic hearts. a** The tSNE analysis segregated a total of 1786 single cardiac cells, obtained from micro-dissected heart regions of primate embryos at 4 and 7 weeks of fetal age, into 13 clusters, including SHF/OFT progenitors (cluster #5). **b** Heatmap image depicting the representative differentially expressed genes in each of the 13 clusters in **a**. **c** Feature plots of the pan-cardiac and first heart filed (FHF)-related genes, as well as the SHF marker genes on the tSNE plots in **a**. **d** Violin plots of the same genes as in **c** in the segregated 13 clusters of the primate embryonic heart-derived single cells. **e** The rankings of the growth factor genes correlated with expression of the SHF-specific gene *ISL1* (left) and the pan-cardiac/FHF-specific gene *NKX2-5* (right) in single-cell RNA-seq data of primate embryonic hearts. The top 8 growth factors are highlighted in red shades, respectively. The corrected *P*-value for each gene was calculated by Guilt-by-Association and correlation analysis[26] (with Pearson correlation coefficient test). Source data are provided as a Source Data file. **f** Atlas of

growth factor expression from multiple heart cell types in developing hearts (i.e., SHF, FHF, CM, pacemaker cell [PM], EC, SMC, and cardiac fibroblasts [CFB]), which was analyzed and defined by the Seurat and Guilt-by-Association and correlation analyses using single-cell RNA-seq data of primate embryonic hearts. In the Seurat program, each cluster (e.g., SHF: cluster #5, FHF: cluster #9, etc. [Supplementary Fig. 2a, b])-specific growth factors were identified, while in the latter, the growth factors correlated with each of the cell type-specific markers (i.e., SHF: *ISL1*, FHF: *NKX2-5*; CM: *TNNT2*; PM: *SHOX2*; EC: *PECAM1*, SMC: *ACTG2*, and CFB: *DCN*) were identified. The red, yellow, blue, and sky-blue colors indicate that each growth factor is ranked within the top 300 (red), top 301–600 (yellow), top 601–1200 (blue), or top 1201–2000 (sky-blue) genes correlated with each of the clusters and/or the cell type-specific markers. Thus, a red color in the chart indicates the strongest association between each of the growth factors and each of the heart cell types.

vasculogenesis via enhancing the function of heart progenitors among the representative 52 growth factors, and generated modRNAs of these 24 growth factors, which involve: (1) the top 12 growth factors that are highly co-expressed in the human ISL1⁺ SHF/OFT progenitor cells (Supplementary Fig. 1e); (2) 6 growth factors (CLEC11A [SCGF], HGF, IGFBP4, NGF, NRG1, and PDGFA) that exhibit high association with CM and/or SMC/EC in Fig. 1f; and (3) 6 growth factors (CSF3 [GCSF], FGF2,

FGF4, FGF17, RLN2, and TNFα) that showed cardiogenic effects in the in vitro hESC-CM differentiation[18,30] pilot studies, using each of recombinant proteins (Supplementary Data 1).

We then tested the potential cardiomyogenic and/or vasculogenic effects of these modRNAs in the in vitro hESC-CM differentiation. In the cardiomyogenic assays, the cells were treated with the modRNA for 5 h on either day 3 or 6 and harvested 3 days later for flow cytometry

analysis to detect differentiated CMs. The gating strategies on flow cytometry experiments are shown in Supplementary Fig. 2b. When modRNAs were added to cell culture on day 3, only PLGF modRNA significantly increased the number of differentiated TNNT2+ CMs on day 6 among the 24 growth factor modRNAs (Fig. 2a, b). The number of totally cultured cells or a cell-proliferation marker Ki67+ cells was not different between the PLGF modRNA-treated cells and control on day 6. Therefore, PLGF appeared to enhance CM differentiation among the hESC-derived cells, rather than promoting CM proliferation. Interestingly, when PLGF or other growth factors' modRNA was added on day 6, the number of differentiated TNNT2+ CMs on day 9 was not increased compared to control (Fig. 2c, d), suggesting PLGF modRNA may promote the early-staged (but not late-staged) heart progenitors to differentiate into more CMs in vitro.

In the vasculogenic assays, the cells were treated with the mod-RNA on day 5 in CM differentiation and employed for flow cytometry analysis to detect differentiated vascular cells on day 8. As expected

from the previous literature[15–17], VEGF-A modRNA significantly increased the number of differentiated PECAM1+ ECs (Fig. 2e, f). Of note, albeit to a lesser degree than VEGF-A, PLGF modRNA also increased the number of differentiated ECs compared to control (Fig. 2e, f). This suggests that PLGF modRNA may promote the late-staged heart progenitors to differentiate into more ECs in vitro. In contrast, no modRNAs increased the number of differentiated PDGFRB+ SMCs in the vasculogenic assays (Fig. 2g, h).

Next, we examined expression patterns of PLGF mRNA in single-cell RNA-seq data of primate embryonic hearts. Intriguingly, in consistency with the findings in the in vitro hESC cardiogenesis assay, PLGF was specifically expressed in the SHF/OFT progenitors (cluster #5) in the early stage and expressed in the ECs (cluster #11) and SMCs (cluster #13) in the late stage (Fig. 3a–c). To corroborate these findings, we then employed immunostaining of human embryonic sectioned hearts. We observed that PLGF+ cells appeared predominantly in the OFT region of the early-staged heart (5.5 weeks of fetal age), often co-expressing an

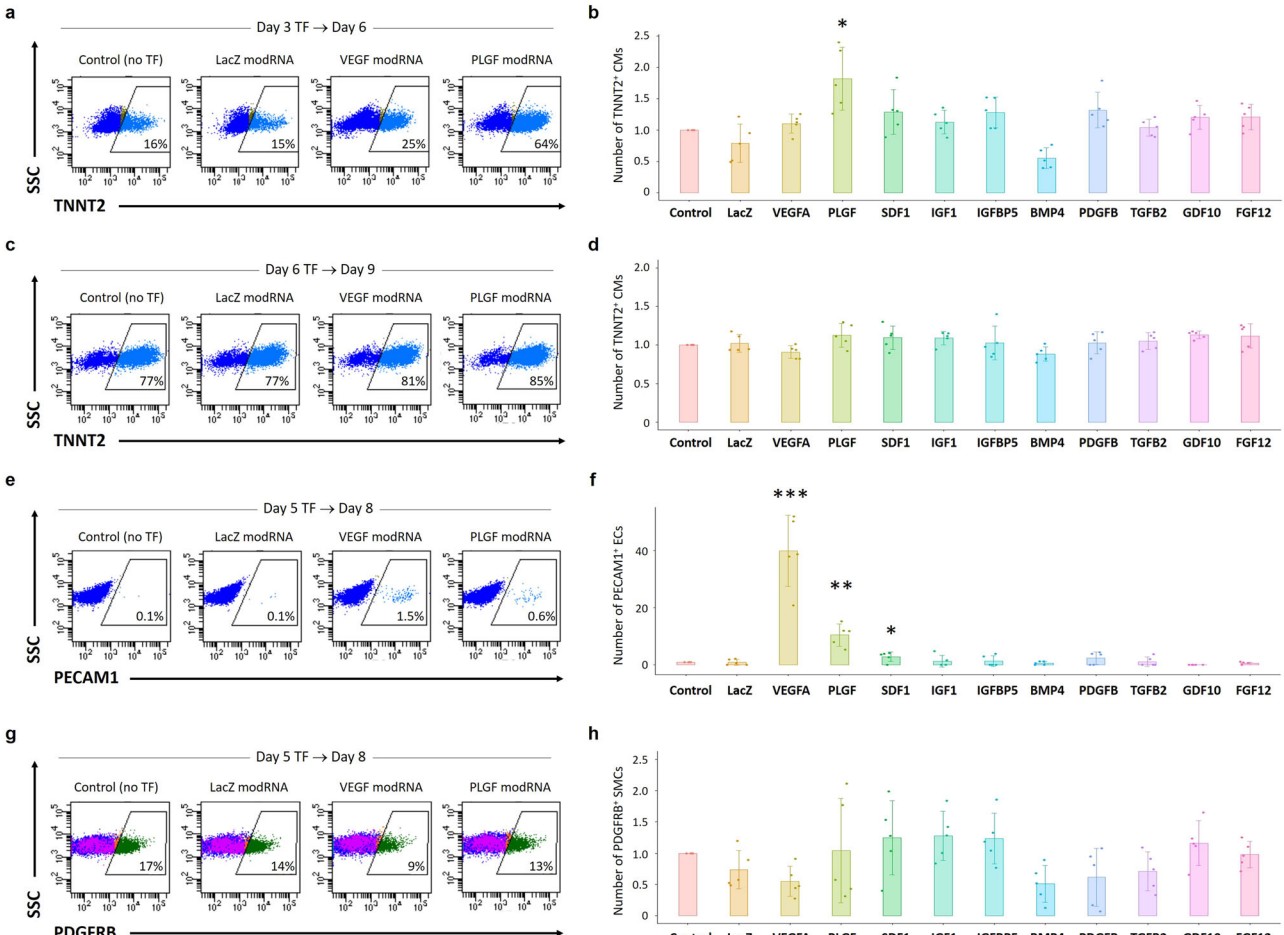

**Fig. 2 | Cardiomyogenic and vasculogenic effects of modRNAs encoding growth factors in in vitro hESC differentiation assays. a**, **b** The cardiomyogenic assays were analyzed on day 6. For the in vitro hESC-CM differentiation[18,30], the cells were treated with each of the 24 selected growth factors' modRNAs for 5 h on day 3 and analyzed for a CM marker TNNT2 by flow cytometry on day 6. The cell numbers of TNNT2+ CMs were then calculated. The panels in **a** show selected representative images on flow cytometry analysis, and the chart in **b** shows relative ratios of the cell numbers of TNNT2+ CMs, obtained with treatment with the representative 10 growth factors' modRNAs, as well as control (no modRNA transfection [TF]) and LacZ modRNA-transfected cells. Of note, only PLGF modRNA significantly increased the number of TNNT2+ CMs on day 6 among all growth factors tested. **c**, **d** The cardiomyogenic assays were analyzed on day 9. The cells were treated with modRNAs encoding each of the 24 selected growth factors for 5 h on day 6 and

analyzed for a CM marker TNNT2 by flow cytometry on day 9. **e**, **f** The vasculogenic assays for ECs analyzed on day 8. In the hESC-CM differentiation, the cells were treated with each of the 24 selected modRNAs encoding growth factors for 5 h on day 5 and analyzed for an EC marker PECAM1 by flow cytometry on day 8. VEGF-A modRNA significantly increased the number of PECAM1+ ECs, as expected. Albeit to a lesser degree than VEGF-A, PLGF modRNA also showed a significant increase in the number of ECs compared to the control. **g**, **h** The vasculogenic assays for SMCs analyzed on day 8. The same cells as in **e** and **f** were also analyzed for an SMC marker PDGFRB by flow cytometry. Data in **b**, **d**, **f**, and **h** are presented as mean ± SD (n = 5 independent experiments). Differences between groups were examined with one-way ANOVA followed by Tukey multiple comparisons test. *P < 0.05, **P < 0.01, and ***P < 0.0001 vs. control. Source data are provided as a Source Data file.

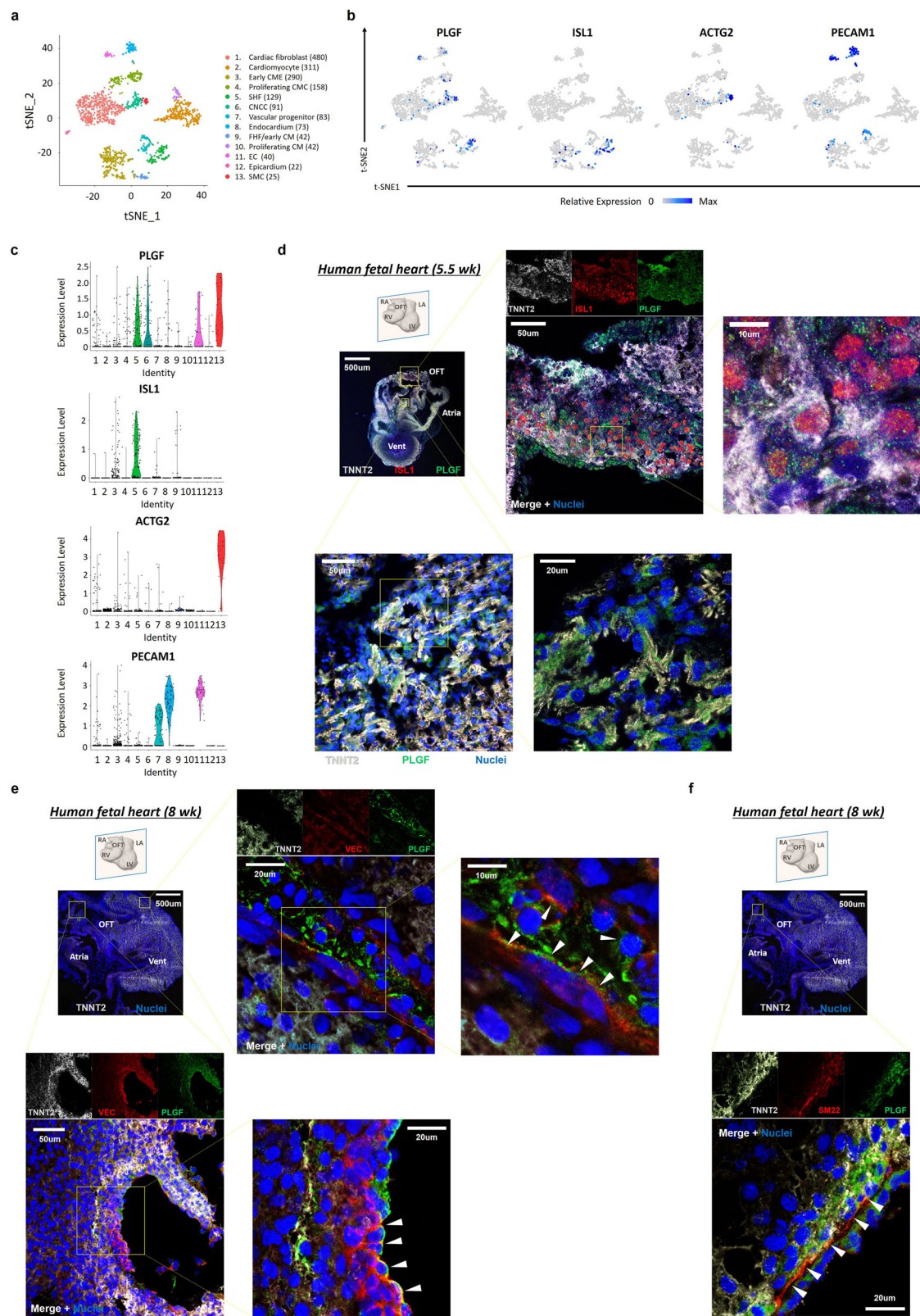

SHF/OFT progenitor marker ISL1 and/or a CM marker TNNT2 (Fig. 3d). In contrast, in the late-staged heart (≥8 weeks), PLGF expression was often seen in VE-cadherin (VEC)⁺ ECs/endocardial cells (Fig. 3e) and SM22⁺ SMCs (Fig. 3f) in the heart, irrespective of their anatomical locations. Collectively, these indicate that PLGF shows biphasic and specific expression patterns in the SHF heart progenitors and CM intermediates at the earlier stage and in the vascular cells at the later stage during heart

development, and may thereby exert a dual effect for both cardiomyogenesis and vasculogenesis.

## PLGF deletion attenuates the induction of both cardiomyocytes and vascular cells

To further explore the functional role of PLGF in human cardiomyogenesis and vasculogenesis, we generated *PLGF*-knockout (*PLGF*-KO)

**Fig. 3 | Distribution of PLGF expression on the human and primate embryonic hearts. a** The 13 cell populations were segregated by the Seurat/tSNE analysis using a total of 1786 single cardiac cells of primate embryonic hearts. **b** Feature plots of *PLGF*, a SHF marker *ISL1*, a SMC marker *ACTG2*, and an EC marker *PECAM1* on the tSNE plots in **a. c** Violin plots of the same genes as in **b**, in the segregated 13 clusters of the primate embryonic heart-derived single cells. **d** Immunohistochemistry of the sectioned human embryonic heart at 5.5 weeks of fetal age. Coronal view. The confocal microscopic images highlight the PLGF+ cells (green) co-expressing an SHF marker ISL1 (red) and/or a CM marker TNNT2 (light gray) in the outflow tract (OFT) region. Vent, ventricle. **e** Immunohistochemistry of the sectioned human

embryonic heart at 8 weeks of fetal age. Coronal view. The confocal microscopic images highlight the PLGF+ ECs (top; arrowheads) in the ventricular wall and the PLGF+ endocardial cells (bottom; arrowheads) in the atrium, both of which co-expressed an endothelial marker VE-cadherin (VEC; red). **f** Immunohistochemistry of the sectioned human embryonic heart at 8 weeks of fetal age. Coronal view. The confocal microscopic images highlight the PLGF+ SMCs, which co-expressed a SMC marker SM22 (arrowheads). Representative images in each of **d**, **e**, and **f** were obtained from the repeated experiments (*n* = 2 [**d**] or 3 [**e** and **f**] biologically independent samples) with similar results.

hESC lines for the loss-of-function experiments. Through CRISPR/Cas9 technology[31], we established the two *PLGF*-KO hESC clones, which had frameshift mutations in the third exon of the *PLGF* gene in both alleles, respectively (Supplementary Fig. 4). During the in vitro hESC-CM differentiation, expression of PLGF protein in wild-type (WT) cells peaked at day 3 and was detectable until day 12 (Supplementary Fig. 5a, b). In contrast, very little or no expression of PLGF protein could be detected in *PLGF*-KO hESC-derived cells during CM differentiation (Supplementary Fig. 5c).

Next, to detect any cardiogenic phenotype in regard to *PLGF* deletion, the *PLGF*-KO hESC clones underwent in vitro CM differentiation using previously published protocols[18,30] and we measured the ratios of cells that were positive for a cell-proliferation marker Ki67[32], a SHF heart progenitor marker ISL1, and a differentiated CM marker TNNT2 on days 6 and 15 by flow cytometry (Fig. 4a–c). The gating strategies are shown in Supplementary Fig. 2b. On day 6 when ISL1 expression reaches a peak, the *PLGF*-KO hESC-derived cells exhibited lower ratios of both ISL1+ and TNNT2+ cells compared to WT control ([%ISL1+] WT $86.0 \pm 7.3\%$ vs. KO $49.4 \pm 22.9\%$, $P < 0.05$; [%TNNT2+] WT $31.5 \pm 7.3\%$ vs. KO $2.7 \pm 3.7\%$, $P < 0.01$), although %Ki67+ did not change between WT and *PLGF*-KO cells (Fig. 4a, c). Likewise, on day 15, the *PLGF*-KO hESC-derived cells exhibited a significantly lower ratio of TNNT2+ cells compared to WT (WT $77.0 \pm 17.7\%$ vs. KO $19.1 \pm 14.6\%$; $P < 0.01$) (Fig. 4b, c). The number of the beating CMs (TNNT2+) generated from *PLGF*-KO hESCs was also much lower than those generated from WT hESCs (WT $3.6 \pm 0.5 \times 10^6$ per well vs. KO $0.9 \pm 0.5 \times 10^6$ per well; $P < 0.01$). These results indicate that *PLGF* would be important for proper induction of CMs from hESCs in vitro.

To evaluate capabilities to differentiate into vascular cells, we then employed the *PLGF*-KO hESC clones in the in vitro SMC[33] and EC[34] differentiation protocols. In line with the previous reports, the meso-dermal lineage-derived vascular SMCs (PDGFRB+) and ECs (VEC+) were obtained from WT hESCs on day 6 in SMC and EC differentiation, occupying around 70% (SMC) and 50% (EC) of the cultured cells, respectively (Fig. 4d, e). Of particular interest, *PLGF*-KO hESC-derived cells exhibited much lower efficacies for induction into both vascular SMCs and ECs ([SMC] WT $72.6 \pm 18.7\%$ vs. KO $10.8 \pm 8.8\%$, $P < 0.01$; [EC] WT $45.5 \pm 7.6\%$ vs. KO $2.8 \pm 0.9\%$, $P < 0.0001$) (Fig. 4d, e). These results strongly support the notion that *PLGF* would be essential for induction into both vascular cells, i.e., SMCs and ECs from hESCs in vitro.

Finally, we performed rescue experiments for the *PLGF*-KO hESC clones using recombinant human PLGF protein (Peprotech) in in vitro differentiation assays. Treatment with 50 or 100 ng/mL of PLGF protein during days 3–7 in CM differentiation and days 4-6 in SMC and EC differentiation improved the efficacies for induction of TNNT2+ CMs, PDGFRB+ SMCs, and VEC+ ECs in *PLGF*-KO hESC-derived cells, respectively (Supplementary Fig. 6).

**PLGF modRNA promotes the upregulation of heart and vasculature development-related genes**

To clarify molecular signatures in human PLGF-associated cardiac development, we analyzed population RNA-seq data obtained from both WT and *PLGF*-KO hESC-derived cells on days 4 and 6 of the hESC-CM differentiation protocol. WT hESC-derived cells that were

transfected with PLGF or LacZ (control) modRNA on day 3 or 5 and then harvested on day 4 or 6, respectively, were also involved in the analysis. Principal component analysis and differential gene expression analysis clearly segregated the four cell groups in a stage-dependent manner (Fig. 5a, b). We then compared directly the transcriptomes between WT and PLGF modRNA-transfected cells, as well as between WT and *PLGF*-KO cells on the same differentiation day with the limma package[35] in R/Bioconductor, and performed the gene set enrichment analysis (GSEA) using the GSEA software (Broad Institute; http://www.gsea-msigdb.org/gsea/) (Fig. 5c–j).

On day 4, the genes upregulated in PLGF modRNA-transfected cells compared to WT cells were enriched for gene ontology (GO) terms such as heart development, muscle tissue development, vasculature development, and mesenchyme development, which contained genes such as *HAND2, MEF2C, HOXA1, HEY1, TBX5, MEIS2, PDGFRA, FGF10, MYOCD, TGFB2*, etc. (Fig. 5c, d; Supplementary Data 2). In contrast, the genes upregulated in WT cells compared to PLGF modRNA-transfected cells on day 4 were enriched for GO terms such as neurogenesis, endoderm differentiation, mesodermal commitment pathway, and ectoderm differentiation, which contained genes such as *SOX2, FOXA2, SOX17, POU5F1, EOMES, LHX1, HHEX, TBXT*, etc. (Fig. 5c, d; Supplementary Data 2). In analogy with the relationship between the PLGF modRNA-transfected and WT cells, the genes upregulated in WT cells compared to *PLGF*-KO cells on day 4 were enriched for GO terms such as heart development, mesenchyme development, muscle structure development, and vasculature development, which contained genes such as *BMP4, NKX2-5, TBX20, TBX5, KDR, APLNR, PDGFRA, HAND1, HAND2, HEY1, MEIS2, MYOCD, TGFB2, PDGFRB*, etc. (Fig. 5e, f; Supplementary Data 2). In contrast, the genes upregulated in *PLGF*-KO cells compared to WT cells on day 4 were enriched for GO terms such as cell migration, apoptotic process, regulation of cell death, and neurogenesis, which contained genes such as *ANXA1, JUN, CFLAR, CCL2, ID1, FOS, RGCC, POU5F1, FOXA1, SOX2, TDGF1*, etc. (Fig. 5e, f; Supplementary Data 2).

On day 6, the genes upregulated in the PLGF modRNA-transfected cell group compared to WT cells were also enriched for GO terms such as heart development, striated muscle contraction, cardiac muscle contraction, and muscle tissue development, which contained genes such as *MEF2C, NKX2-5, MYOCD, ACTN2, TNNT2, MYH6, MYH7, ACTA2, PLN*, etc. (Fig. 5g, h; Supplementary Data 3). The upregulation of these sarcomere proteins and cardiac/smooth muscle actins indicated a more differentiated status of the in vitro cardiac cells on day 6 than on day 4. In contrast, the genes upregulated in WT cells compared to PLGF modRNA-transfected cells on day 6 were enriched for GO terms such as epithelium development, regulation of locomotion, morphogenesis of an epithelium and mesodermal commitment pathway, which contained genes such as *GATA3, FOXA2, SOX17, POU5F1, ZFP42, TBX3, HHEX, BMP4*, etc. (Fig. 5g, h; Supplementary Data 3). Again, in analogy with the relationship between the PLGF modRNA-transfected and WT cells, the genes upregulated in WT cells compared to *PLGF*-KO cells on day 6 were enriched for GO terms such as heart development, muscle structure development, circulatory system development, and regulation of heart contraction, which contained genes such as *NKX2-5, MEF2C, TPM1, TNNT2, MYH6, TBX5, ACTN2, PLN, CACNA1C, MYOCD, ACTA1*, etc. (Fig. 5i, j; Supplementary Data 3). In contrast, the genes

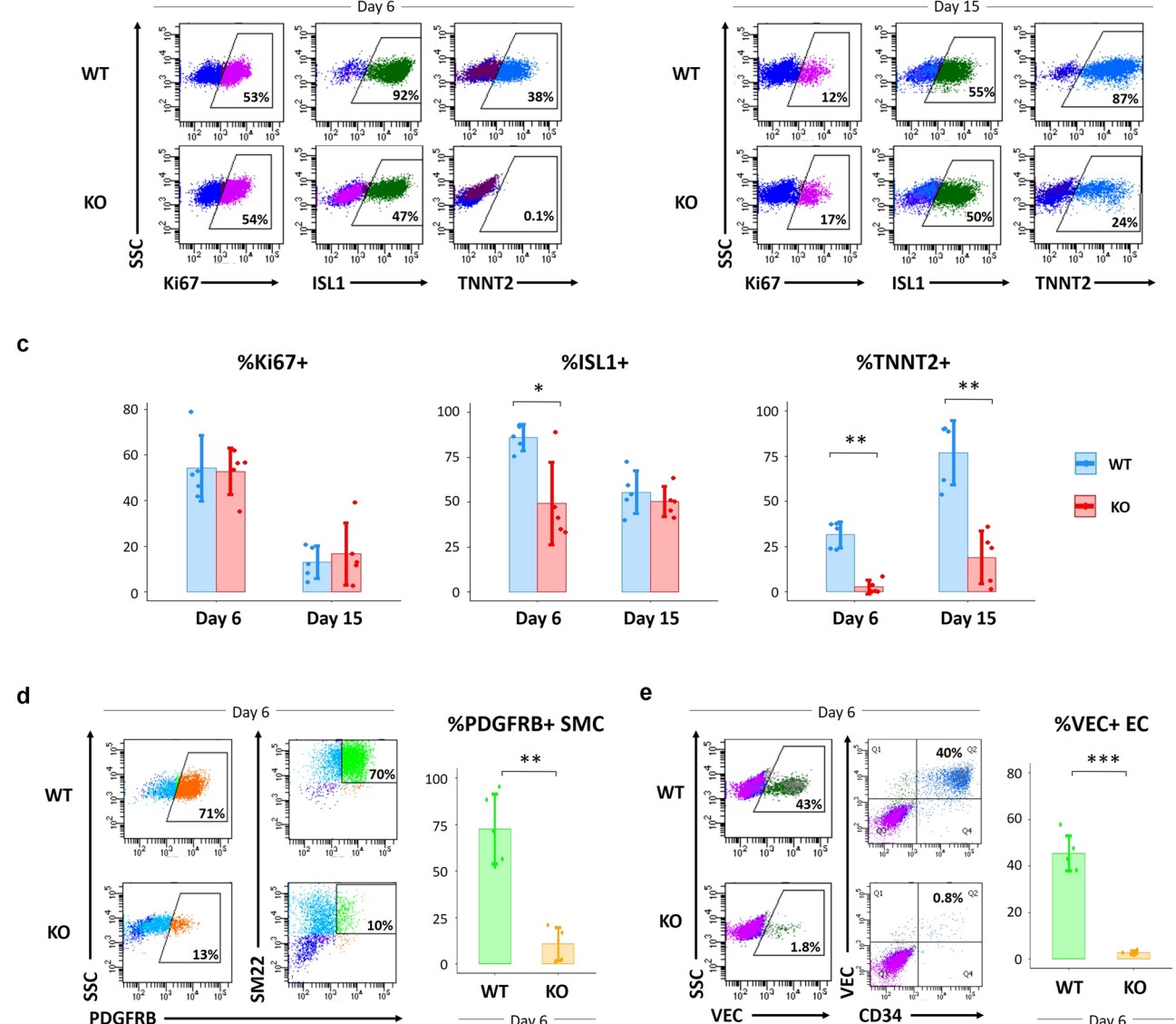

**Fig. 4 | Impacts of PLGF deletion in in vitro hESC differentiation of CMs, SMCs, and ECs. a, b** Representative images on flow cytometry analysis showing the ratios of a cell-proliferation marker Ki67[+] (left), a SHF/heart progenitor marker ISL1[+] (middle), and a differentiated CM marker TNNT2[+] (right) in WT (top) and *PLGF*-KO (bottom) cells at days 6 (**a**) and 15 (**b**) in hESC-CM differentiation[18,30]. **c** Statistical data of the ratios of %Ki67[+] (left), %ISL1[+] (middle), and %TNNT2[+] (right) in **a** and **b**. **d** Flow cytometry analysis and statistical data showing the ratios of vascular SMCs (PDGFRB[+]) at day 6 in hESC-SMC differentiation[33] of WT and *PLGF*-KO hESCs. **e** Flow cytometry analysis and statistical data showing the ratios of vascular ECs (VE-cadherin [VEC][+]) at day 6 in hESC-EC differentiation[34] of WT and *PLGF*-KO hESCs. Data in **c**, **d**, and **e** are presented as mean ± SD (*n* = 5 independent experiments). Differences between groups were examined with one-way ANOVA followed by Tukey multiple comparisons test. \**P* < 0.05, \*\**P* < 0.01, and \*\*\**P* < 0.0001. Source data are provided as a Source Data file.

upregulated in *PLGF*-KO cells compared to WT cells on day 6 were enriched for GO terms such as ectoderm differentiation, embryonic morphogenesis, negative regulation of multicellular organismal process, and negative regulation of cell population proliferation, which contained genes such as *HNF4A, MLXIPL, CFLAR, OVOL2, OTX1, ZIC3, FOXA2, SOX17*, etc. (Fig. 5i, j; Supplementary Data 3).

Collectively, these transcriptional analyses revealed that PLGF modRNA would be a positive enhancer for both heart and vasculature development (i.e., cardiomyogenesis and vasculogenesis) when administered at the heart progenitor stage during in vitro hESC cardiogenesis, which was consistent with the findings in the in vitro differentiation assays (Fig. 2). In fact, expression of a great number of the cardiomyogenesis drivers, such as typical transcription factors (e.g., *HAND2, MEF2C, MYOCD*, etc.) and chemical mediators (e.g., *WNT, FGF*, etc.), was significantly upregulated by treatment with PLGF modRNA while reversely downregulated by *PLGF* deletion (Supplementary Fig. 7a). This enhancement of the cardiomyogenesis programs was more predominant when the differentiating cells were treated with PLGF modRNA earlier (day 3 > 5). On the other hand, a number of vasculogenesis drivers (e.g, *ETV1, APLNR, ANGPT1, VCAM1*, etc.) were also significantly upregulated by treatment with PLGF modRNA while reversely downregulated by *PLGF* deletion, which was more predominant when the differentiating cells were treated with PLGF modRNA later (day 3 < 5) (Supplementary Fig. 7b). These results suggest that PLGF would activate both the cardiomyogenesis and vasculogenesis programs in a stage-dependent fashion. In contrast, *PLGF* deletion would induce cell migration, lower cell viabilities, cell apoptosis, and/or shifting of the differentiating cell trajectories towards the ectoderm (e.g., neurons, epithelium) or endoderm lineages (e.g., hepatocytes) from the

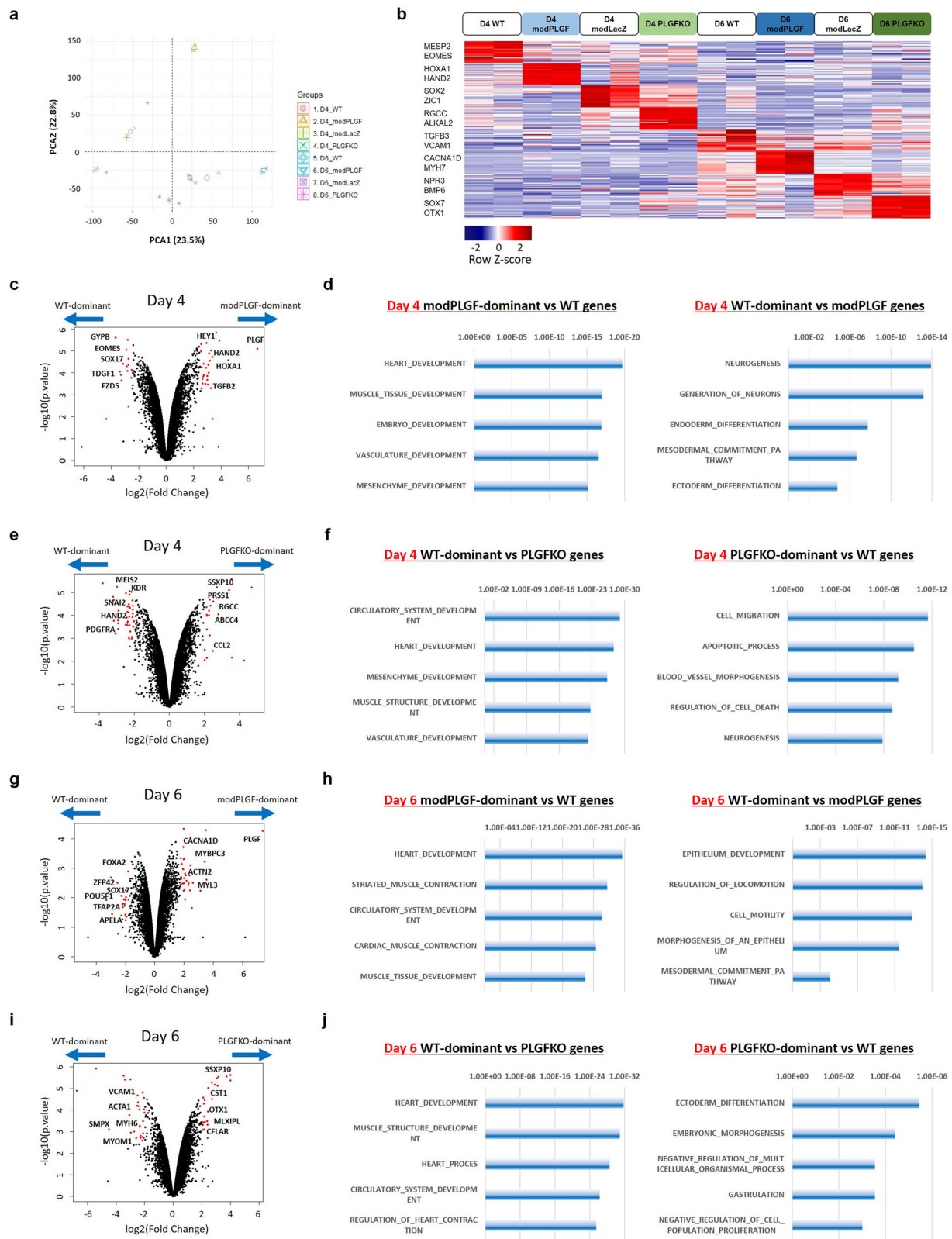

originally destined mesoderm lineages during the in vitro hESC-CM differentiation.

## PLGF modRNA exerts in vivo cardiomyogenic and vasculogenic effects

To elucidate the effects of PLGF in the context of in vivo differentiation of human heart progenitors (HPs) for cardiac muscle formation, we

transplanted hESC-derived HPs that were transfected with or without PLGF modRNA into kidney capsules of immunocompromised mice[19]. We employed the same in vitro CM differentiation protocol[18], and cardiac differentiating cells at day 3 were electroporated with 5 μg of GFP (control) or PLGF modRNA and harvested at day 5 (termed early HPs). Separately, cardiac differentiating cells at day 5 were electroporated with 5 μg of GFP or PLGF modRNA and harvested at day 6

**Fig. 5 | Population RNA-seq analyses compared between WT cells and PLGF overexpressing or PLGF-KO cells during hESC-CM differentiation. a** The principal component analysis using the population RNA-seq data of WT, modRNA (LacZ or PLGF)-transfected WT, and *PLGF*-KO cells harvested at days 4 and 6 in hESC-CM differentiation. ModRNA transfection was conducted 24 h before cell harvesting (i.e., at day 3 or 5). **b** Differential gene expression analysis of the 8 cell groups in **a**. Heatmap image depicting the representative differentially expressed genes (partly listed in the left column) in each of the 8 groups. **c, e, g, i** Volcano plots visualizing differentially expressed gene analysis with the limma package[35] between WT and PLGF modRNA-transfected cells (**c** [day 4] and **g** [day 6]), as well as between WT and *PLGF*-KO cells (**e** [day 4] and **i** [day 6]) in hESC-CM differentiation, respectively. For each gene, the average difference ($\log_2$[Fold change]) between the cell groups on the same day was plotted against the power to discriminate between groups (-$\log_{10}$[p.value]), in which p.values were obtained from a two-tailed unpaired *t*-test. Top-scoring genes for both metrics are indicated as red dots, and representative differentially expressed genes' names are labeled. **d, h** The gene set enrichment analysis (GSEA) was performed using the top 250 WT or PLGF modRNA-transfected cells-enriched genes with the GSEA software (Broad Institute; http://www.gsea-msigdb.org/gsea/). Bar graphs showing the representative gene ontology (GO) terms specific to WT (right) or PLGF modRNA-transfected cells (left) at days 4 (**d**) and 6 (**h**), respectively. **f, j** The GSEA was performed using the top 250 WT or *PLGF*-KO cells-enriched genes with the GSEA software. Bar graphs showing the representative GO terms specific to WT (left) or *PLGF*-KO cells (right) at days 4 (**f**) and 6 (**j**), respectively. Source data are provided as a Source Data file.

(termed late HPs). From a pilot study, we observed that injection of the combined early and late HPs could generate bigger heart muscle grafts on a kidney capsule than injection of only the single early or late HPs (relative increase ratios of the graft areas: $1.95 \pm 0.64$; $P < 0.01$).

We then injected the following groups of cells into murine kidney capsules, respectively: (1) 1.5 million of intact early HPs plus 1.5 million of intact late HPs, which were harvested at day 5 and 6 without any modRNA transfection (No TF control; 3 million cells in total); (2) 1.5 million of the GFP modRNA-transfected early HPs plus 1.5 million of the GFP modRNA-transfected late HPs (GFP control; 3 million cells in total); (3) 1.5 million of the PLGF modRNA-transfected early HPs plus 1.5 million of the PLGF modRNA-transfected late HPs (3 million cells in total). The engrafted kidneys were harvested at 1 month after surgery (Fig. 6a). Interestingly, the PLGF modRNA-transfected HP-engrafted kidneys were heavier in weight than controls (Fig. 6b), likely reflecting the bigger sizes of their generated cardiac muscle grafts in vivo.

In histological analyses, as expected, the PLGF modRNA-transfected HP-engrafted cardiac grafts presented with larger cross-sectional areas and TNNT2$^+$ areas when compared to controls (Fig. 6c, d), suggesting that the PLGF modRNA-transfected HPs would differentiate into more CMs than No TF and GFP controls upon in vivo transplantation. Similarly, areas that were positive for the CM maturation marker MLC2V were larger in the PLGF modRNA-transfected HP-engrafted group when compared to controls, indicating that PLGF modRNA also promoted cardiac maturation in the in vivo muscle grafts (Fig. 6e, f). Next, we examined the expression of a proliferation marker Ki67 in the grafts and found the number of Ki67$^+$ cells in the PLGF modRNA-transfected HP-engrafted group was higher than those in No TF and GFP controls (Fig. 6e, f), suggesting that PLGF modRNA could enhance proliferation of the transplanted HP-derived cardiac cells upon in vivo transplantation, likely leading to the larger sizes of the grafts. We then assessed to what extent vascular structures formed within the grafts through histological examination of an EC marker (VEC) and an SMC marker ($\alpha$-smooth muscle actin [$\alpha$SMA]), respectively. Of note, the PLGF modRNA-transfected HP-engrafted cardiac grafts exhibited significantly higher vascular density (VEC$^+$) than controls (Fig. 6g, h), which most likely contributed to the bigger sizes of the in vivo generated cardiac grafts. Overall, these findings demonstrated the in vivo cardiomyogenic and vasculogenic effects of PLGF modRNA upon HP transplantation.

## PLGF is a target of both EOMES and SOX17

To clarify molecular machinery in human PLGF-associated cardiomyogenesis and vasculogenesis, we explored a previously unrecognized transcription network behind PLGF. We focused on an early cardiogenic TCF EOMES[20] and a vasculogenic TCF SOX17[21], as both genes were significantly upregulated in WT cells compared to PLGF modRNA-transfected cells, as well as upregulated in *PLGF*-KO cells compared to WT cells, on day 4 (EOMES) and on days 4 and 6 (SOX17) (Fig. 5). This evidence may imply that both genes function upstream of *PLGF* during the in vitro cardiac differentiation. Through the TCF motif

analysis with the Jaspar database (http://jaspar.genereg.net/) and the MatInspector software (Genomatix, http://www.genomatix.de/), we found the putative EOMES-binding motif sites (5'-(AG)GTGTGA-3') on the *PLGF* promoter region, which is 1934, 1408, 790, and 724 bp upstream of the transcription start site (TSS) of the human *PLGF* gene (Fig. 7a, top). We also found the putative SOX17-binding motif sites (5'-(C/T)ATTGT(C/G)−3') on the *PLGF* promoter region, which are 3901, 1638, 1542, and 1384 bp upstream of the TSS of the human *PLGF* gene (Fig. 7a, bottom).

Next, chromatin immunoprecipitation (ChIP) assays with antibodies specific to EOMES and SOX17 (Abcam) were performed using extracts derived from WT hESC-derived cells on days 3 and 6 in the in vitro CM differentiation. We first confirmed successful and specific protein immunoprecipitation with anti-EOMES and anti-SOX17 antibodies by western blotting analysis. We then found that recruitment of EOMES protein was significantly augmented onto one of the putative EOMES-binding motifs on the *PLGF* promoter (−1934 bp upstream from the TSS) on day 3 and 6 (day 3 > 6) (Fig. 7b). On the other hand, recruitment of SOX17 protein was augmented onto one of the putative SOX17-binding motifs on the *PLGF* promoter (−3901 bp upstream from the TSS) on day 3 and onto all of the four putative SOX17-binding motifs on day 6 (Fig. 7c). These suggest that the identified EOMES-PLGF transcriptional interaction would function at the earlier embryonic stage, while the identified SOX17-PLGF transcriptional interaction would function at and following the early embryonic stage.

In consistency with the findings on the ChIP assays, the Guild-by-Association and correlation analysis using the primate embryonic heart single-cell RNA-seq data (Fig. 1) revealed that *SOX17* expression was highly positively co-related to *PLGF* expression (corrected *P*-value: 1.1E-4), especially in the late stage (7 weeks of fetal age), and *SOX17* was strongly expressed in the EC population (cluster #11), showing overlap with *PLGF* expression (Supplementary Fig. 8a−c). Since the early cardiogenic TCF *EOMES* had little or no expression in both human and primate embryonic heart single-cell RNA-seq datasets, we also analyzed the previously obtained single-cell RNA-seq data of in vitro hESC-derived cells during CM differentiation (Supplementary Fig. 8d)[6,18]. The 366 high-quality individual cells were segregated into 6 clusters, in which we identified that cluster #2 was occupied by cells on day 3 in CM differentiation and expressed *EOMES* globally (thus, considered as early cardiac precursors) as well as *PLGF*, as shown in the feature plots on the tSNE plots (Supplementary Fig. 8e, f). These also imply that the putative EOMES-PLGF and SOX17-PLGF transcriptional interactions would be sequentially vital in a time-dependent fashion, i.e., for cardiomyogenesis in the early embryonic stage (EOMES-PLGF) and for vasculogenesis in the middle/late embryonic stage (SOX17-PLGF), respectively (Fig. 7d).

Lastly, we investigated expression patterns of *PLGF* and *FLT1* (*VEGFR1*), a PLGF-specific receptor gene, in the single-cell RNA-seq datasets of the primate embryonic hearts and the in vitro hESC-derived cardiac cells (Supplementary Fig. 8g−j). Of interest, the in vitro early cardiac precursors (cluster #2 in Supplementary Fig. 8d) often co-expressed *PLGF* and *FLT1*, indicating the autocrine signaling and a cell-autonomous role of PLGF in these cells (Supplementary Fig. 8i, j). In the

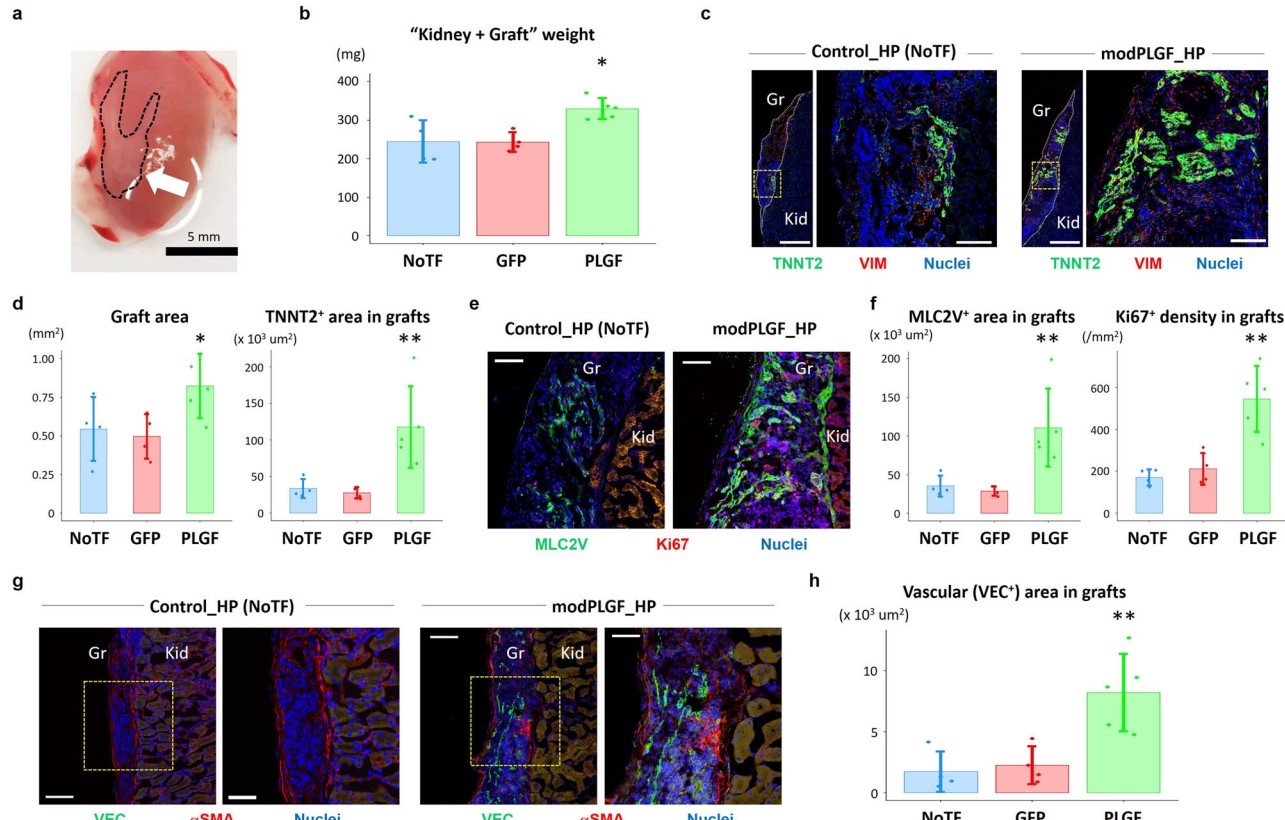

**Fig. 6 | In vivo cardiomyogenic and vasculogenic effects of PLGF modRNA-enhanced HPs. a** In vivo human heart progenitor (HP)-derived cardiac muscle grafts on murine kidney capsules (arrow) were generated by transplantation of hESC-derived HPs. **b** Comparison of the weights of the non-transfected (NoTF) HP-, GFP modRNA (modGFP)-transfected HP-, and PLGF modRNA (modPLGF)-transfected HP-engrafted kidneys. **c** Immunohistochemistry of the sectioned human HP-derived cardiac muscle grafts on murine kidney capsules, generated by NoTF HPs (left) and modPLGF-transfected HPs (right). The grafts (Gr) were indicated by white dotted lines in the left images, respectively. The right image in each is the enlarged one of a yellow square in the left image, respectively. Scale bars, 500 μm (left in each) and 100 μm (right in each). Kid, kidney; VIM, vimentin. **d** Quantitative data of the entire graft areas (left) and TNNT2+ areas in grafts (right) in the three groups, i.e., NoTF-, modGFP-, and modPLGF-HPs. Quantitative analyses were conducted using ImageJ/FIJI software (NIH, USA). **e** Immunohistochemistry of the sectioned human HP-derived cardiac muscle grafts on murine kidney capsules generated by

NoTF HPs (left) and modPLGF-transfected HPs (right), highlighting a CM maturation marker MLC2V and a proliferation marker Ki67. Scale bars, 100 μm. **f** Quantitative data of MLC2V+ areas in grafts (left) and Ki67+ density in grafts (right) in the three groups. **g** Immunohistochemistry of the sectioned human HP-derived cardiac muscle grafts on murine kidney capsules generated by NoTF HPs (left) and modPLGF-transfected HPs (right), highlighting an EC marker VE-cadherin (VEC) and a SMC marker α-smooth muscle actin (αSMA). The right image in each is the enlarged one of a yellow square in the left image, respectively. Scale bars, 100 μm (left in each) and 50 μm (right in each). **h** Quantitative data of VEC+ areas in grafts in the three groups. Data in **b**, **d**, **f**, and **h** are presented as mean ± SD (n = 4–5 biologically independent samples). Differences between groups were examined with one-way ANOVA followed by Tukey multiple comparisons test. *P < 0.05 and **P < 0.01 between modPLGF-transfected HP-engrafted kidneys vs. NoTF HP- or modGFP-transfected HP-engrafted kidneys. Source data are provided as a Source Data file.

primate heart cells at 4 weeks when the SHF cells (cluster #5 in Supplementary Fig. 8a) expressed *PLGF* specifically, while some of the SHF cells co-expressed *FLT1*, many of the vascular progenitor cells (cluster #7) expressed *FLT1* but not *PLGF* (Supplementary Fig. 8g, h). This suggests that at the early stage in vivo, the SHF-derived PLGF would play a dual role, i.e., an autocrine role in the SHF and a paracrine role in affecting the vascular progenitor cells for promoting intercellular vascular differentiation (Fig. 7e). In the primate heart cells at 7 weeks when the ECs and SMCs (cluster #11 and #13 in Supplementary Fig. 8a) expressed *PLGF* specifically, while ECs, as well as endocardium (cluster #8), expressed *FLT1* strongly, SMCs showed little or no expression of *FLT1* (Supplementary Fig. 8g, h). Thus, it is suggested that at the late stage in vivo, the EC-derived PLGF would play a cell-autonomous (autocrine) role, whereas the SMC-derived PLGF would play a paracrine role in affecting ECs and endocardium in promoting endothelial differentiation (Fig. 7e).

Collectively, we have newly identified the previously unrecognized gene regulatory network, involving interactions between *EOMES* and *PLGF*, as well as *SOX17* and *PLGF*, and the stage-dependent

autocrine and paracrine roles of PLGF, all of which would contribute to human PLGF-mediated cardiomyogenesis and vasculogenesis during heart development.

## Discussion

A wide variety of cardiogenic paracrine mediators including growth factors, cytokines/chemokines, and secreted proteins play critical roles during cardiogenesis[8,9]. Here, we focused on the roles of growth factors, which are expressed spatiotemporally and function on specific heart cell types during cardiogenesis. We established atlases of growth factors uniquely expressed in multiple heart cell types using single-cell RNA-seq analyses of human and primate embryonic hearts (Fig. 1; Supplementary Figs. 1 and 3). Importantly, we discovered a previously unknown role of PLGF, which exhibited the biphasic and specific expression pattern on the SHF progenitors and CM intermediates at the early embryonic stage and on vascular cells (i.e., SMCs and ECs) at the middle/late stage during cardiogenesis (Fig. 3). Using modRNA technologies[13,14], we tested and confirmed the dual effect of PLGF modRNA promoting cardiomyogenesis and vasculogenesis in vitro

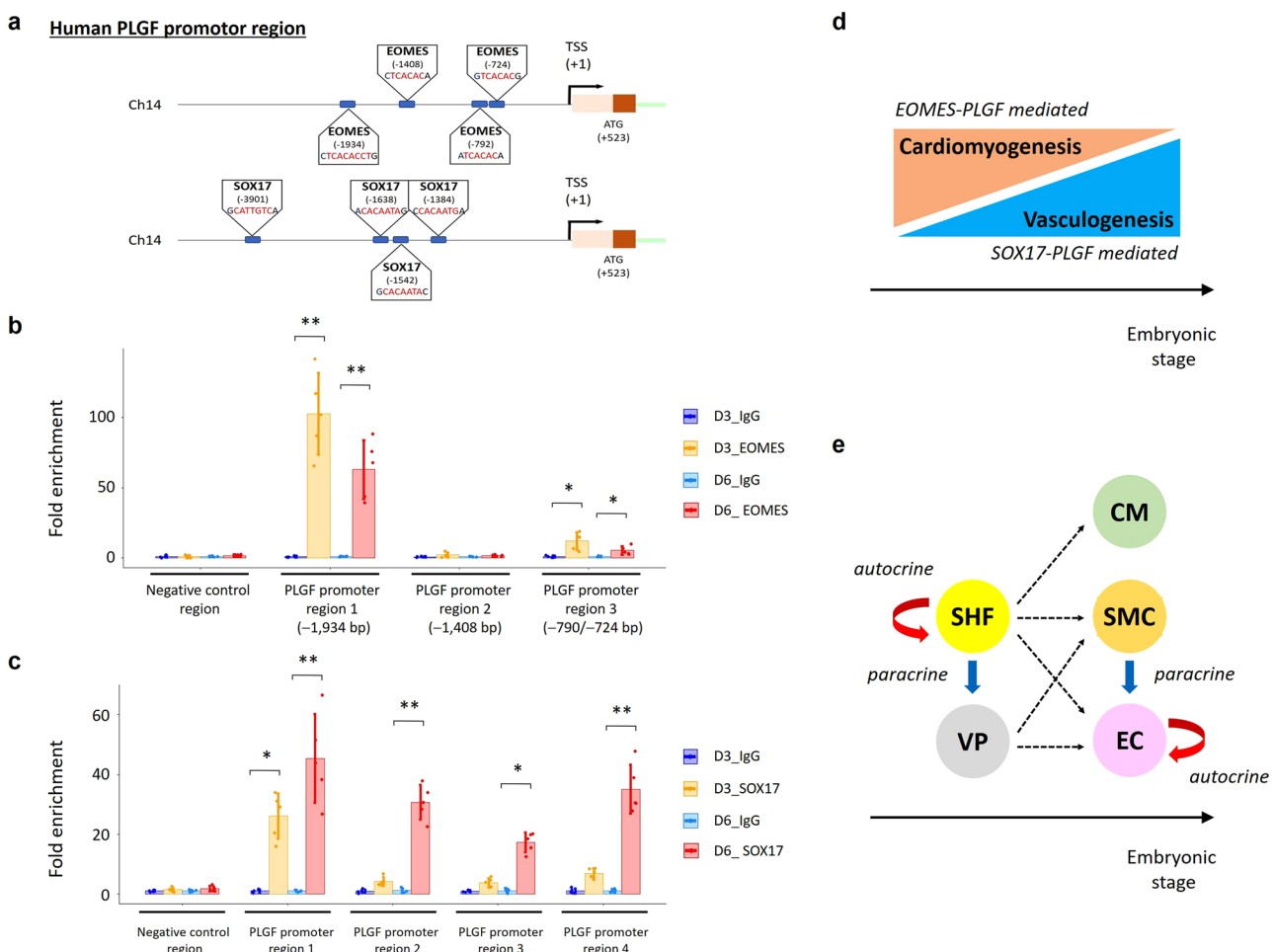

**Fig. 7 | Chromatin immunoprecipitation assays highlight the binding sites of EOMES and SOX17 on the PLGF promoter region. a** Schematic showing the putative binding sites of a cardiogenic transcription factor EOMES (5′-[AG] GTGTGA-3′; top) and a vasculogenic transcription factor SOX17 (5′-[C/T]ATTGT[C/G]−3′; bottom) on the human *PLGF* promoter region. **b** Chromatin immunoprecipitation (ChIP) assays demonstrated that recruitment of EOMES protein onto one of the putative EOMES-binding motif sites of the human *PLGF* promoter (−1934 bp upstream from the transcription start site [TSS]) was significantly augmented at days 3 (D3) and 6 (D6) in CM differentiation of WT hESCs. The degree of fold enrichment was larger at day 3 than at day 6. Albeit to a lesser degree, recruitment of EOMES was also detected onto another putative EOMES-binding motif site (−790/−724 bp upstream from the TSS) at D3 and D6. **c** The ChIP assays demonstrated that recruitment of SOX17 protein was significantly augmented onto one of the putative SOX17-binding motif sites of the human *PLGF* promoter (−3901 bp

upstream from the TSS) at D3 and onto all of the four putative SOX17-binding motif sites (−3901, −1638, −1542, and −1384 bp upstream from the TSS) at D6. Data in **b** and **c** are presented as mean ± SD (*n* = 5 independent experiments). Differences between groups were examined with one-way ANOVA followed by Tukey multiple comparisons test. *$P < 0.01$ and **$P < 0.0001$ vs. IgG (negative control). Source data are provided as a Source Data file. **d** Schematic highlighting the EOMES-PLGF-mediated cardiomyogenesis in the early embryonic stage and the SOX17-PLGF-mediated vasculogenesis in the middle/late embryonic stage. **e** Schematic highlighting the stage-dependent PLGF's roles in cardiogenesis. In the early stage, the second heart field (SHF) heart progenitors play a dual role, such as an autocrine role in the SHF and a paracrine role to function to the vascular progenitors (VP). In the late stage, smooth muscle cells (SMCs) play a paracrine role in functioning to endothelial cells (ECs), while ECs play an autocrine role. Black dashed arrows indicate putative differentiation paths. CM cardiomyocyte.

(Fig. 2) and in vivo (Fig. 6). In contrast, genetic ablation of the *PLGF* gene significantly attenuated the generation of ECs and SMCs, as well as CMs, in in vitro hESC differentiation (Fig. 4). These in vitro and in vivo cardiomyogenic and vasculogenic effects of PLGF were well supported by analyses of the population RNA-seq data, obtained from in vitro hESC-derived cardiac cells with gain or loss of the *PLGF* gene's function (Fig. 5). Further, through machinery analysis, we identified previously unrecognized interactions between a cardiogenic TCF EOMES[20] and PLGF at the earlier embryonic stage, as well as a vasculogenic TCF SOX17[21] and PLGF at the later developing stage (Fig. 7), which may explain the time-dependent regulation of the PLGF's dual function in the autocrine and paracrine forms for cardiomyogenesis and vasculogenesis.

PLGF, a selective ligand of FLT1/VEGFR1, is a member of the VEGF family and is mainly expressed in the placenta and other tissues such

as skeletal muscle, lung, and heart[10,36]. Previous studies have shown that the PLGF-FLT1 axis functioned endothelial cell growth and angio-/vasculogenesis[37,38], and that exogenous administration of recombinant human PLGF protein could reduce infarct size and improve cardiac function by enhancing angiogenesis and arteriogenesis after acute myocardial infarction in mice[39] as well as in chronic myocardial ischemia in pigs[11]. Beyond the angiogenic properties, PLGF has also been reported to exert its cardioprotective effects via the reduction of oxidative stress and CM apoptosis in murine myocardial ischemia/reperfusion injury models[12]. Interestingly, Accornero et al. further revealed that PLGF functioned as a stress-response paracrine factor and regulated cardiac adaptation and hypertrophy by providing protective trophic effects to ECs, fibroblasts, and CMs during pressure overload in mice[40]. However, because the potential role of PLGF in cardiogenesis had not yet been

determined, its authentic functions in developing hearts and heart progenitors were still not clear.

Through the latest technologies involving single-cell RNA-seq and modRNA, we attempted to address the potential roles of cardiogenic growth factors in human cardiogenesis. Here we have demonstrated that beyond the previously reported angiogenic and cardioprotective capabilities, PLGF drives and enhances the function of the early cardiac progenitors such as ISL1[+] SHF and/or EOMES[+] heart precursors and possesses its cardiomyogenic properties in addition to its vasculogenic properties. The loss-of-function tests using *PLGF*-KO hESCs further emphasized the indispensable roles of PLGF in in vitro hESC differentiation into CMs, SMCs, and ECs. To dissect the molecular pathways that were enhanced following transfection with PLGF modRNA or the disruption of such pathways by *PLGF* ablation, we performed the population RNA-seq analysis using *PLGF*-KO and WT hESC-derived cardiac cells with or without transfection with PLGF modRNA. In consistency with the previous studies[12,41], *PLGF* deletion led to activation of the apoptotic process, as represented by upregulation of *CFLAR*[42], *CCL2*[43], and *RGCC*[44]. In addition, interestingly, *PLGF* deletion also induced shifting of the differentiating cell trajectories from the originally destined mesoderm lineages towards the ectoderm lineages (e.g., neurons, epithelium) as represented by upregulation of *OTX1*[45], *OVOL2*[46], and *SOX2*[47], or the endoderm lineages (e.g., hepatocytes) as represented by upregulation of *FOXA1*[48], *HNF4A*[49] and *MLXIPL*[50]. Unexpectedly, this indicates an important role of PLGF in the mesodermal lineage's commitment and specification. Furthermore, PLGF appears to regulate not only vasculature development but also heart development, directly and/or indirectly, as represented by the upregulation of multiple cardiogenic TCFs, such as *NKX2-5*, *MEF2C*, *HAND1*, and *TBX5*[51], and sarcomere proteins. Although we showed that an early cardiogenic TCF EOMES[20] would function upstream of PLGF and likely switch on the PLGF's cardiomyogenic properties, which was followed by upregulation of the cardiogenic driver genes (Supplementary Fig. 7), elucidating more detailed molecular mechanisms downstream of PLGF for cardiomyogenesis require further investigation. Nonetheless, these findings suggest more pleiotropic effects of PLGF in the heart and cardiogenesis, than previously thought.

In this study, we utilized modRNA technologies[13] for gain-of-function tests. Unlike plasmid DNA and viral vectors often used in gene therapy, modRNA offers efficient, dose-dependent, transient protein expression and low innate immunogenicity, as previous studies have demonstrated the feasibility of targeted mRNA delivery for vaccination in humans[14]. In the field of cardiovascular biology and medicine, the modRNA of an angiogenic mediator VEGF-A has been shown to exert therapeutic effects by enhancing angiogenesis in animal and human ischemic disease models[15,16]. Notably, these beneficial angiogenic effects by VEGF-A modRNA were not accompanied by the adverse side effects (e.g., formation of angioma-like structures, vascular leakage, edema, etc.) that had been often observed in the conventional (plasmid or virus) gene therapy of VEGF-A[52], which is most likely due to the modRNA's advantageous pharmacokinetics to induce transient and pulse-like but non-sustained expression of protein in vivo[17]. Of particular interest, we and our collaborators have found that the cell (e.g., mesenchymal stem cell)-based transfer of transfected modRNA resulted in much higher protein expression and thereby exhibited more potent biological effects in vitro and in vivo, compared to naked modRNA administration alone[53]. Nevertheless, considering the identified properties of PLGF to promote both cardiomyogenesis and vasculogenesis, the administration of PLGF modRNA combined with or without cell therapies may serve as an ideal treatment strategy for therapeutic heart regeneration and structural repair for deficit hearts. In fact, we observed the in vivo effects of PLGF modRNA with HPs for the generation of more mature and larger heart muscle grafts with more developed vasculature on murine kidney capsules. A kidney capsule model is reported to be a sophisticated chimeric cell transplantation model to investigate in vivo cellular differentiation[19], as the histological appearances and degrees in differentiation and proliferation of the human heart progenitors after transplanted into both kidney capsules and hearts in immunocompromised mice were very similar and well correlated mutually between the two organs[54,55]. Yet future work is warranted to validate authentic therapeutic effects of the PLGF modRNA-transfected HPs to promote heart regeneration and repair for deficit hearts using large animal models. Another limitation in this study would be that some clusters in the single-cell analyses of the human and primate embryonic hearts contained relatively small numbers of cells, due to a technical reason (e.g., manually picking the dissociated human cardiac cells) and/or the rarity of those clusters (e.g., heart progenitors), as this could be a potential limitation on the interpretation of the obtained data.

In conclusion, our study has identified a dual function and sequential autocrine and paracrine roles of PLGF for cardiomyogenesis and vasculogenesis during heart development, thereby suggesting a therapeutic potential of PLGF modRNA for heart disease. Thus, our study paves the way for further studies of developing novel therapeutic strategies combined with modRNA and cardiac progenitors/CMs for heart regeneration and repair in the future.

## Methods

### Ethics statement
The study was performed in accordance with the Declaration of Helsinki and the guidelines from Directive 2010/63/EU, and all the protocols including handling of human samples and animal works were approved by the institutional review board at Karolinska Institutet (KI) with ethical permission numbers (Dnr 2015/1369-31/2 [human]; N277/14 [primate]; and N227-14 [mouse]). On animal works, to reduce potential pain, suffering, or distress, fentanyl was pre-operatively given at 0.03 mg/kg intraperitoneally as a painkiller. Anesthesia was induced by isoflurane during surgery, and temgesic (buprenorphine) was post-operatively given at 0.1–0.2 mg/kg intraperitoneally or subcutaneously for pain relief. We carefully monitored the animals during and after surgery and checked the distress and pain scores by following the institutional rules established by Karolinska Institutet. All the processes were also observed and checked by the veterinarians.

### Isolation of human and primate embryonic heart-derived single cells
For the analysis of human developing hearts, we utilized our previously published single-cell RNA-seq dataset[6]. In fact, embryonic and fetal materials collection was carried out at Karolinska University Hospital in Huddinge (Sweden) in the study. Only after the patient decides to undergo an abortion for any reason, the medical staff at the Gynecology department inform her and her partner (or closest relatives) about the possibility of donating the embryo/fetus for only research purposes with documents describing the kinds of research that would be performed. After giving their informed consent for the donation of the embryo or fetus, the patient underwent a surgical abortion, and the aborted material was dissected under sterile conditions. Finally, a total of 458 individual cardiac cells derived from micro-dissected heart regions (i.e., OFT, atria, and ventricles) of human embryonic/fetal hearts ($n = 7$) at 4.5–10 weeks of the gestation stages were participated in the single-cell RNA-seq dataset[6].

For the analysis of primate (macaque fascicularis) developing hearts, we combined previously reported single-cell RNA-seq data of the early-staged (4 weeks) primate hearts[56] with our obtained single-cell RNA-seq data of the late-staged (7 weeks) primate heart. For the latter, we first obtained primate blastocysts that were developed from zygotes that were generated by in vitro fertilization as described previously[57,58], from the facility of Yunnan Key Laboratory of Primate Biomedical Research in China. Embryos (blastocysts) were transferred into the oviducts of the matched female monkey recipients

(aged 5–8 years) with proper hormone levels of β-estradiol and progesterone at Astrid Fagræus laboratory in Karolinska Institutet. The pregnancy was diagnosed by ultrasonography at 2–3 weeks after embryo transfer, and when terminating a pregnancy, a cesarean section was performed. The harvested heart was micro-dissected into OFT, RV, LV, and atria, and the divided heart regions were cut into small pieces and dissociated into single cells with solutions containing collagenase type II (200-400 U/ml; Worthington)[6,18]. After staining the dissociated cells with DAPI (Thermo Fisher Scientific), DAPI-negative (live) single cells were sorted into 384-well plates containing cell lysis buffer, customized for the Smart-seq2 approach[59], using a fluorescence-activated cell sorter (FACSARIA III; BD Biosciences).

### Single-cell RNA library construction and RNA sequencing
cDNA libraries of the sorted single cardiac cells were prepared with the Smart-seq2 approach composed of the sequential steps such as denature, reverse transcription, PCR pre-amplification, PCR purification, c, enrichment PCR, re-PCR purification, quality check, and library pooling[59]. The pooled libraries were sequenced at 125 bp paired-end or 50 bp single-end to a mean read depth of around 700,000 totally aligned reads per cell on the Hiseq 2500 or 4000 instrument (Illumina).

### Single-cell RNA-sequencing data analysis
Raw reads were pre-processed with the sequence-grooming tools FASTQC and Cutadapt and followed by sequence alignment with the STAR aligner and Samtools onto human (hg38) or primate (macFas5) genome reference with default settings (unique mapping rate: around 75–85%)[26]. Mapped gene counts were carried out with HTSeq, and transcript levels were quantified for each transcript as fragments or read per kilobase of transcript per million mapped reads (FPKM or RPKM). The average number of expressed genes (FPKM/RPKM ≥ 1) was approximately 7000–8000 per cell, and the average number of counts was approximately 15-20 per gene. After filtration of low-expression genes and poor-quality cells, depth-normalization of the filtered cells' reads was conducted using edgeR or DESeq programs, implemented in R/Bioconductor. To overview the entire single-cell transcriptomics to be analyzed, principal component analysis (PCA), hierarchical clustering, and diffusion map dimensionality reduction were performed[60]. To cluster the analyzed single cells, dimensionality reduction methods such as two-dimensional t-distributed stochastic neighbor embedding (tSNE) and uniform manifold approximation and projection (UMAP) were performed using the Seurat program[27,61]. Differential expression analysis of the defined clusters was conducted using edgeR, Limma, and Seurat programs[6].

### Human ESC culture
The hESC line H9 was purchased from WiCell Research Institute and maintained on feeder-free and 0.3 mg/ml Matrigel (BD Biosciences)-coated plates in mTeSR1 medium (STEMCELL Technologies), according to manufacturers' instructions. Cells were fed daily and passaged every 4–5 days with Accutase (STEMCELL Technologies). Media was supplemented with 5 µM ROCK inhibitor Y-27632 (Tocris) for 24 h after splitting.

### In vitro CM differentiation
Cardiac-directed differentiation was performed using a well-established hESC-CM differentiation protocol based on Wnt signaling modulation[18,30]. Briefly, hESCs were dissociated into single cells with Accutase and seeded onto Matrigel-coated 12-well plates at 750,000–1.2 million cells per well in mTeSR1 supplemented with 5 µM Y-27632 for 24 h (day 2). At day 0, cells were treated with 12 µM GSK-3β inhibitor CHIR99021 (Sigma) in RPMI medium supplemented with B27 minus insulin (RPMI/B27-ins; Thermo Fisher Scientific) for 24 h. At day 1, the medium was replaced with fresh RPMI/B27-ins. At day 3, half

of the medium in each well was changed to RPMI/B27-ins supplemented with 5 µM Wnt inhibitor IWP-2 (Tocris), all of which was replaced with fresh RPMI/B27-ins at day 5. At day 7, the medium was switched to fresh RPMI medium with B27 supplement (RPMI/B27). Thereafter, the medium was replaced with fresh RPMI/B27 every other day. Beating CMs start to appear in the culture typically on day 8–9. Robust and broad spontaneous contractions are observed from day 10–12 onward.

### In vitro SMC differentiation
In vitro mesodermal lineage-derived vascular SMC differentiation from hESCs was performed using a previously reported protocol with minor modification[33]. Briefly, dissociated hESCs were seeded onto Matrigel-coated 6-well plates at 400,000 cells per well in mTeSR1 supplemented with 5 µM Y-27632 for 24 h (day 0). At day 1, the medium was replaced with N2/B27 medium (1:1 of DMEM/F12 and Neurobasal medium [Gibco] with B27 and N2 supplements [Thermo Fisher Scientific] and 0.1% β-mercaptoethanol) supplemented with 8 µM CHIR99021 and 25 ng/ml human BMP4 (R&D), which was maintained for 3 days. On days 4 and 5, the medium was replaced with fresh N2/B27 medium supplemented with 10 ng/ml PDGF-BB (Peprotech) and 2 ng/ml Activin-A (Peprotech). Typically, mesodermal vascular SMCs (platelet-derived growth factor receptor-β [PDGFRB]⁺SM22⁺) show up from day 6 onward.

### In vitro EC differentiation
In vitro vascular EC differentiation from hESCs was performed using a previously reported protocol[34]. Briefly, dissociated hESCs were seeded onto Matrigel-coated 6-well plates at 250,000 cells per well in mTeSR1 supplemented with 5 µM Y-27632 for 24 h (day 0). On day 1, the medium was replaced with N2/B27 medium supplemented with 10 µM CHIR99021 and 20 ng/ml human BMP4 (R&D), which was maintained for 3 days. At days 4 and 5, the medium was replaced with the EC induction medium, i.e., StemPro-34 medium (Gibco) containing StemPro-34 supplement (Gibco), 1% L-glutamine (Invitrogen), 50 ng/ml VEGF-A (Peprotech), and 10 µM of a Notch inhibitor DAPT (Sigma). Typically, vascular ECs (PECAM1⁺CDH5⁺) show up from day 6 onward.

### Construction of plasmids for the growth factor ModRNA library
Construction of DNA plasmids to generate in vitro transcription (IVT)-produced mRNA was performed as previously described[62]. Briefly, DNA constructs containing genes of interest were ordered from Dharmacon or Addgene. Primers were designed that spanned the open reading frame (ORF) of the genes of interest. PCR reactions were performed using HiFi HotStart DNA polymerase (KAPA Biosystems) per the manufacturer's instructions. Amplified ORFs were treated using T4 polynucleotide kinase (New England Biolabs). PCR products were then purified using QIAquick spin columns (Qiagen). Blunt ligations between the PCR products and a template plasmid (pORF-in) that incorporates generic 5′ and 3′ untranslated regions (UTRs) and a poly-A tail, as previously described[62], were performed using T4 ligase (New England Biolabs). Newly ligated plasmids were propagated using the TOPO TA cloning kit (ThermoFisher Scientific) and verified by Sanger sequencing in the KI gene facility (Stockholm, Sweden).

### ModRNA synthesis
ModRNAs of the selected cardiogenic and vasculogenic growth factors were produced as previously described[17]. Briefly, mRNA was synthesized using T7 RNA polymerase-mediated IVT from a linearized DNA template containing the target gene's ORF, flanking 5′ and 3′ UTRs and a poly-A tail. A Cap1 structure was enzymatically added to the 5′ end to produce the final mRNA. Uridine was completely substituted with N1-methylpseudouridine to reduce potential immunostimulatory activity and to improve protein expression relative to unmodified mRNA. ModRNA was then purified using the Ambion MEGAclear kit and

treated with Antarctic phosphatase (New England Biolabs) for 30 min at 37 °C to remove residual 5′-phosphates. All modRNAs were re-purified and quantified using Nanodrop (Thermo Scientific). After purification, modRNA was resuspended in MEGAclear buffer at 1 μg/μL concentrations and frozen at −80 °C for future use.

## ModRNA transfection

ModRNA transfections were carried out using RNAiMAX (Thermo Fisher Scientific). For single-well RNAiMAX transfections in 12-well plates, every 5 μL of modRNA (at 1 μg/μL) was diluted in 45 μL Opti-MEM (Invitrogen) and separately 12.5 μL RNAiMAX was diluted in 37.5 μL Opti-MEM. These components were then combined and left to incubate for 15 min at room temperature before being dispensed in the cell culture media. ModRNA transfections were performed in freshly added RPMI media (Thermo Fisher Scientific) and left to incubate for 5 h, after which time the media was again changed to 2 mL fresh RPMI. Cells were harvested 72 h after transfection for flow cytometry analysis, or 24–48 h after transfection for in vitro population RNA-sequencing and in vivo transplantation experiments, as noted below.

## Validation assays of ModRNAs in in vitro CM differentiation

To assess the potential cardiogenic and vasculogenic effects of mod-RNA on the selected growth factors, modRNAs were added to cardiac progenitor cells at pertinent time points of the in vitro CM differentiation protocol[18,30]. For the cardiogenic assays, 5 μg/well of selected modRNA candidates were added to the cells for 5 h on either day 3 or 6, and the cells were collected for flow cytometry analysis 3 days later, on day 6 or 9 of the differentiation protocol, respectively. For the vasculogenic assays, 5 μg/well of selected modRNA candidates were added for 5 h on day 5 of the differentiation protocol. On day 8 (72 h later), the cells were collected for flow cytometry analysis.

## Flow cytometry analysis

Flow cytometry analysis was performed for cells at days 6, 9, and 15 in CM differentiation and at day 6 in SMC and EC differentiation. Cells were dissociated into single cells with Accutase for 5–10 min, washed in PBS, and blocked for 30 min in fluorescence-activated cell sorting (FACS) buffer (1% bovine serum albumin and 10% horse serum in PBS) at 4 °C. Staining for cell surface antigens was first performed in the vasculogenesis assay of CM differentiation and in SMC and EC differentiation for 30 min at 4 °C using the following primary antibodies: anti-PDGFRB-PE (a SMC marker; BD Biosciences), anti-PECAM1-FITC (an EC marker; BD Biosciences), anti-VE-cadherin-PE (an EC marker; BD Biosciences) and anti-CD34-APC (an endothelial progenitor marker; BD Biosciences). Cells were then fixed with 4% paraformaldehyde, permeabilized, blocked, and stained for intracellular antigens for 30 min at room temperature using the following primary antibodies: anti-Ki67-FITC (a proliferative marker; BD Biosciences), anti-ISL1-PE (a cardiac progenitor marker; BD Biosciences), anti-TNNT2-APC (a CM marker; Miltenyi Biotec), and anti-SM22 (a SMC marker; Abcam) followed by staining with an Alexa-Fluor 647-conjugated secondary antibody (BD Biosciences) for 15 min at 4 °C. Flow cytometry analysis was then conducted with a flow cytometer (FACSARIA III) and FACS Diva software (Beckton Dickinson). The gating strategies on flow cytometry experiments are shown in Supplementary Fig. 2. Flow cytometry data were analyzed with FACS Diva and FlowJo software (Tree Star). All the primary antibodies used in flow cytometry analyses are listed in Supplementary Table 1.

## In vivo transplantation of ModRNA-transfected heart progenitors into murine kidney capsules

To generate the human HP-derived cardiac muscle grafts and assess the in vivo effects of modRNA on the selected growth factors, we used a heart progenitor-modRNA hybrid transplantation system. For this, we first transfected modRNA of the selected growth factors into hESC-derived heart progenitors on day 3 or 5 of the in vitro CM differentiation protocol[18,30]. For transfection of modRNA, the cells were incubated with RNAiMAX (Thermo Fisher Scientific) for 5 h at 37 °C. 5 μg of modRNA was placed per well of 12-well plates including hESC-derived heart progenitors. ModRNA-transfected cells were harvested 24–48 h after transfection and stored in liquid nitrogen for future use.

For in vivo experiments, three million hESC-derived heart progenitors transfected with or without modRNA were collected and transplanted under a kidney capsule of the immunodeficient male NOD.Cg-Prkdc[SCID] Il2rg[tm1Wjl]/SzJ (NSG) mice (10–12 weeks old; Charles River)[63]. We followed a previously described protocol for kidney transplantation under analgesia and inhalation anesthesia with 2–2.5% isoflurane[19]. Engrafted kidneys were harvested at 1 month after surgery for histological analysis.

## Immunostaining

Human embryonic hearts and murine-engrafted kidneys were snap-frozen and cryosectioned at 10-μm thickness. The sectioned samples were fixed with 4% paraformaldehyde, permeabilized in PBS with 0.1% saponin, and blocked in PBS with 1% bovine serum albumin and 10% horse serum. The specimens were then stained with primary anti-bodies at 4 °C overnight, followed by three washes with PBS and incubation with Alexa-Fluor 488-, 594-, and/or Alexa-Fluor 647-conjugated secondary antibodies (Molecular Probes) specific to the appropriate species for 60 min at room temperature. After three washes with PBS, nuclei were counterstained with DAPI (Sigma), or the slides were mounted in Vectashield Mounting Medium with DAPI (Vector Laboratories). All images were obtained using a Zeiss 710 confocal microscope and its imaging system. All the primary anti-bodies used at immunostaining are listed in Supplementary Table 1. Quantitative analyses were conducted using ImageJ/FIJI software (NIH, USA).

## Generation of *PLGF*-KO hESCs by CRISPR-Cas9

The following sequence was selected as a single guide-RNA (sgRNA) which targets the third exon of the human *PLGF* gene locus and has minimal off-target activity, using the CHOPCHOP software (http://chopchop.cbu.uib.no/): 5′-CGTGTCCGAGTACCCCAGCG(AGG)−3′. The sgRNA was cloned into a bicistronic expression vector pX459 expressing S. pyogenes Cas9[64], following a previously published protocol[31]. Human ESCs (H9) were transiently co-transfected with pX459-sgRNA and a plasmid encoding a puromycin resistance gene using the Human Stem Cell Nucleofector Kit (Lonza), according to the manufacturer's instructions. After drug selection with 0.5 μg/ml puromycin for 2 days, single clones were obtained by re-plating transfected cell pools at low density (5000 cells per dish) on Matrigel-coated 10 cm dishes. Cells were allowed to grow for 6–10 days until single colonies were big enough to pick and transferred to a 96-well plate. Monoclonal cell lines were then expanded, and genomic DNAs of each clone were isolated using the GeneJET Genomic DNA Purification Kit (Thermo Fisher Scientific). The CRISPR/Cas9-mediated gene edition on the human *PLGF* gene locus in each clone was confirmed by Sanger sequencing (Euro-fins) and aligned to a wild-type (WT) sequence of the target region with SnapGene (GSL Biotech LLC) (Extended Data Fig. 3). To confirm the absence of PLGF protein expression in *PLGF*-KO hESC-derived cells in CM differentiation, western blotting analysis was performed with an anti-PLGF antibody (Abcam), as noted below.

## In vitro population RNA-sequencing and data analysis

WT and *PLGF*-KO hESC-derived cells, as well as PLGF or control (LacZ) modRNA-transfected (at one day before collection) hESC-derived cells, were harvested on days 4 and 6 of the CM differentiation protocol. Their cDNA libraries were generated using Illumina TrueSeq mRNA (poly-A selection) kits. Each library was sequenced at 150 bp paired-end on an Illumina NovaSeq 6000 S4 instrument to a depth of

2–4 × 10$^7$ reads. The quality of the fastq-format sequenced data was assessed using FASTQC, and raw reads were further trimmed and aligned onto human genome reference (hg38) using Cutadapt and STAR. Transcript levels were quantified as FPKM. Further normalization and differential expression analysis were conducted using edgeR and Limma programs on R/Bioconductor[6]. The gene set enrichment analysis (GSEA) was performed on the GSEA software v4.1.0 (Broad institute).

## Western blotting

Total cellular protein was extracted from cultured cells using RIPA lysis buffer (Sigma) with a protease and phosphatase inhibitor cocktail (Thermo Scientific). Protein concentration was determined with a BCA protein assay kit (Thermo Scientific). 20 µg of protein lysates were loaded on 4–12% Bis-Tris gel (Thermo Scientific), separated by electrophoresis, and transferred onto a 0.2 µm nitrocellulose membrane with the Trans-Blot Turbo system (Biorad). The membranes were subsequently blocked in 5% skim milk in TBS-T (TBS with 0.1% Tween20) for 30 min at room temperature and incubated with primary antibodies in TBS-T containing 5% bovine serum albumin overnight at 4 °C. The used primary antibodies were as follows: anti-β-actin-HRP (Cell signaling technology, 5125S); anti-PLGF (Abcam, ab9542); anti-EOMES (Abcam, ab23345); and anti-SOX17 (Abcam, ab224637) (Supplementary Table 1). After washing with TBS-T, membranes were incubated with anti-mouse IgG or anti-rabbit IgG secondary antibodies conjugated with HRP (1:2000) in TBS-T containing 5% bovine serum albumin for 1 h at room temperature. After washing, membranes were incubated with Super-Signal West Femto Chemiluminescent Substrate (Thermo Scientific) for 1–5 min and imaged on a Chemidoc (Biorad). Image analysis and quantification were performed on Image Lab software version 6.1 (Biorad). The signal intensity was normalized to the expression of a loading control protein (β-actin) and translated to relative values.

## Chromatin immunoprecipitation (ChIP)

For ChIP experiments, the nuclei from WT and *PLGF*-KO hESC-derived cells on days 3 and 6 of the CM differentiation protocol were prepared, and chromatin digestion was performed using SimpleChIP Plus Enzymatic Chromatin IP Kit (Cell Signaling Technology), according to manufacturer's instructions. The lysates were immunoprecipitated with normal rabbit IgG, anti-EOMES antibody (Abcam, ab216870), or anti-SOX17 antibody (Abcam, ab224637) (Supplementary Table 1) at 4 °C overnight. Immune complexes were incubated with Protein G Magnetic Beads for 2 h at 4 °C with rotation. After washing and eluting chromatin from the magnetic beads, protein-DNA cross-linking was reversed in 5 M NaCl with 40 mg proteinase K by overnight incubation at 65 °C. After a purification process, the precipitated DNA was amplified for fragments of the putative EOMES or SOX17-binding motif sites ([EOMES] 5′-(AG)GTGTGA-3′; [SOX17] 5′-(C/T)ATTGT(C/G)−3′) on the *PLGF* promoter by real-time PCR for 40 cycles on a 7500 Fast Real-Time PCR System (Applied Biosystems) under standard manufacturer's conditions. Data is presented as fold enrichment compared with the IgG negative control. The PCR primers (listed in Supplementary Table 2) were used for the detection of the putative EOMES-binding regions of 1934, 1408, 790, and 724 bp upstream, or for the detection of the putative SOX17-binding regions of 3901, 1638, 1542, and 1384 bp upstream from the transcription start site (TSS) of the human *PLGF* gene, respectively.

## Statistical analysis

Data are presented as mean ± SD. Differences between groups were examined with one-way ANOVA followed by Tukey post-hoc multiple comparisons test. Statistical significance is defined as $P < 0.05$. All bioinformatics analyses were performed using R/Bioconductor, as described above.

## Reporting summary

Further information on research design is available in the Nature Portfolio Reporting Summary linked to this article.

## Data availability

The RNA-seq data reported in this paper have been deposited in the Sequence Read under accession numbers PRJNA510181 and PRJNA983851. Source data are provided in this paper.

## Code availability

The code used in this paper for computational analyses is available upon request.

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

## Acknowledgements

We would like to thank the National Genomics Infrastructure (NGI) Sweden and the Eukaryotic Single Cell Genomics Facility (ESCG) at Science for Life Laboratory for support of RNA sequencing and generating raw sequencing data. We would like to thank Professors Niu and Ji and their laboratories' members at the facility of Yunnan Key Laboratory of Primate Biomedical Research in China for providing with primate blastocysts for embryo transfer experiments. We would like to thank Federica Santoro, Mats Spångberg, and Ran Yang for their help with the primate experiments. pSpCas9(BB)-2A-GFP (PX458) (Addgene plasmid # 48138) and pSpCas9(BB)-2A-Puro (PX459) V2.0 (Addgene plasmid # 62988) were gifts from Feng Zhang. This work was supported by research grants: to K.R.C. from the Swedish Research Council Research collaboration grant China-Sweden (Dnr: 539-2013-7002); the Swedish Research Council Distinguished Professor Grant (Dnr: 541-2013-8351); and the Knut and Alice Wallenberg Foundation (Dnr: 2013.0028); and to M.S. from the Swedish Research Council (Dnr: 2019-01359); the Swedish Heart and Lung Foundation (Dnr: 20150421 and 20190380); and Karolinska Institutet (Strategic Research Area Stem Cells and Regenerative Medicine 2020).

## Author contributions

N.W. contributed to research conception and design, performed in vitro experiments, generation of the modRNA library, collected and assembly of data, and data analysis and interpretation, and wrote the manuscript. C.Z. performed in vitro experiments, generation of PLGF-KO hESC lines, and in vivo experiments. T.H. performed in vitro experiments and the collection and assembly of data. Y.X. performed in vivo experiments. X.H. performed histology works. E.R., J.S., and N.G.B. assisted in vitro experiments and bench works. M.L.L. performed in vivo experiments relating to the primate works. K.R.C. provided research conception and design and financial support, and reviewed all the works and manuscript. M.S. provided research conception and design and financial support, performed in vitro and in vivo experiments and data analysis and interpretation including computational analysis, wrote the manuscript, and reviewed all the works and manuscript.

## Funding

## Competing interests

The authors declare the following competing interests that K.R.C. is a past co-founder of Moderna Therapeutics and is currently a member of the board of directors of eTheRNA. All other authors declare no competing interests.
