## [Peer Review File · Nature Communications]

Placental Growth Factor Exerts a Dual Function for
Cardiomyogenesis and Vasculogenesis during Heart
DevelopmentREVIEWER COMMENTS

Reviewer #1 (Remarks to the Author):

With the aim to identify paracrine mediators that possess the capability to promote cardiac regeneration, the authors focused on growth factors and cytokines that regulate the differentiation and proliferation of cardiac progenitor cells into cardiomyocytes and/or vascular cells during development. This study has identified PLGF, a member of the VEGF family as an important player during cardiogenesis, making use of existing single cell sequencing databases and confirmed their observations in non-human primate embryonic hearts, a function not described to date.

This manuscript opens new venues to understand heart development and the potential for future cardiac regenerative therapies. There are however several questions that remain.

1. From the 52 growth factors, the authors selected 24 to generate modRNA from. From the manuscript, the rationale why these 24 is not clear and might need further explanation.
2. When PLGF modRNA was given 3 days after the initiation of CM differentiation, the number of TNNT2 expression cells increased. One explanation is indeed enhanced differentiation, another explanation can be increased proliferation so more cells can differentiate and/or differences in modRNA uptake as not all cells will be in the same differentiation and cell cycle state.
3. To determine the vasculogenic differentiation, the modRNAs were given to the cells 5 days after initiation of CM differentiation. Why day 5 and what is the percentage of EC/SMC differentiation in the day 6 treatment shown in fig 2c,d?
4. Although very valuable and precious, it is not clear from figure 3e which cells do express PLGF. This reviewer is not convinced that ECs and pericytes are positive, it almost reflects a less mature CM.
5. Comparing 3d and 3e, PLGF is very punctuated @w5.5, is it located in vesicles within the cell?
6. In figure 5 the authors show the RNA-seq analysis of hESC transfected with modRNA. Cell transfected with the LacZ RNA have a distinct profile from the WT cells at D4. This is a bit worrisome. Would one not expect more overlap, and how does this influence the conclusions drawn?
7. Figure 6 nicely shows the in vivo differentiation potential of the cardiac progenitors with or without PLGF. Although a well conducted experiment and nice results, the expression and presence of PLGF is not confirmed. Is the ligand still present and which cell types express PLGF.
8. Analysing the PLGF promoter, the authors identify several EOMES and SOX17 binding sites. They show that depending on the day post differentiation initiation, more SOX17 binding sites are used. Do these two transcription factor compete for promoter binding and what drives the binding of SOX17 to all 4 putative binding sites?
9. Since PLGF signals via FLT-1/VEGFR1, is there co-expression or does PLGF influence intercellular differentiation?
10. Although not reported to be cardiomyogenic, PLGF has been reported to improve cardiac function. Has a potential cardiogenic effect been ignored in these studies?
11. In extended data figure 4, the authors treated PLGF-KO cells with PLGF to rescue the phenotype. Would one not expect that adding PLGF to the WT cells would improve CM differentiation as well?
12. In figure 1, the authors show the different populations present within the human and primate embryonic heart. When analysing the data shown in figure 1a,b, I do get the impression that the clusters are quite far separate from each other in 1a and from the genes shown in 1b, it is not very evident what cell population each cluster represent. In 1c, would one not expect that the Isl-1+ and LRG5+ cells to be more concentrated in one cluster (not necessarily the same)?
13. In figure 1g, another panel of genes is depicted to identify the clusters. This does make comparing the data in this plot with 1b complicated. Can the authors provide a more unified representation?
14. In figure 1k specific growth factors are identified per cell cluster. As these factors can only exert an effect if receptors are also present, can the authors indicate if they are present within the same cell cluster or in a neighbouring one.

Reviewer #2 (Remarks to the Author):

Witman and colleagues demonstrate that placental growth factor (PLGF) promotes cardiomyocyte and endothelial cell differentiation in vitro in human ES cells and infer that such a mechanism drives mammalian cardiac development.

Although PLGF has been shown to promote angiogenesis, it has received little attention in the context of heart development and, as such, there are some novel findings. However, these have not been thoroughly investigated.

The starting point for the study was scRNA-seq from human and primate embryonic hearts. PLGF was identified among the growth factors inferred to putatively promote cardiogenesis and was prioritised for investigation as the only factor (in modRNA form) to promote both cardiogenesis and vasculogenesis. PLGF is expressed in second heart field and outflow tract progenitors at early stages and in endothelial and mural cells at later stages. Subsequent experiments validate the proposed roles in vitro by examining the effect of PLGF modRNA on hESC differentiation and in vivo by transplantation of hESC-derived progenitors into murine kidney capsules. Finally, the authors show ChIP data to suggest that PLGF expression is controlled by EOMES and SOX17 binding within the PLGF promoter.

The mechanism proposed is novel but it has not been investigated thoroughly and the models and methodology is sub-optimal.

1. The scRNA-seq is based on 458 human cardiac cells, which divide into 10 distinct clusters, and 1,789 primate heart cells, with 12 clusters. The authors should indicate the number of cells in each cluster. The most relevant clusters appear to comprise fairly small numbers of cells. This might not be so concerning if not for the highly heterogeneous levels of expression within each cluster, as shown e.g. *Isl1*, *Hand1* on the feature plots in Fig.1.

2. It is unclear why the authors describe EC markers "PECAM1 and CDH5, clusters 8 and 11, to be endocardium and ECs, respectively. Extended Fig 1 shows there is little difference in expression of these genes in cluster 11, as expected. Surely, *NPR3* should be highlighted as a marker of endocardium/ cluster 8? In this instance, I think it is an inaccurate description in the text, rather than inappropriate cluster annotation.

3. In contrast to point 2, I have queries over the cell type annotation of some clusters. For example, there is minimal overlap between *NGFR* and *MYH11*, both highlighted as markers of cluster 6 – this is not altogether surprising as neural crest cells would not be expected to express mature smooth muscle markers. As well as the low cell numbers and extent of transcriptional heterogeneity within clusters, the poor annotation of clusters is a concern. Given the prominence of the scRNAseq data in establishing the central hypothesis of the paper, these data need to be more accurately analysed.

4. The mechanism that the authors propose to be important is not explicitly explained. PLGF is expressed in cardiac progenitors as well as in cardiomyocyte, smooth muscle and endothelial derivatives. They detect a gain of function by ModRNA transfection and can rescue PLGF KO differentiation with recombinant PLGF protein. Thus, it is unclear whether the authors are proposing a

paracrine role (which cell types secrete PLGF to induce progenitor differentiation?) or a cell-autonomous role – autocrine signalling?

5. I do not understand the rationale for selecting a kidney capsule transplantation model to investigate *in vivo* differentiation of PLGF-treated progenitors. It seems an entirely artificial cellular environment in which to investigate cardiovascular differentiation. Moreover, no explanation is offered as to why early and late progenitors are combined in the assay. This may relate to the developmental mechanism that the authors envisage but it has not been explained. Given the discussion around therapeutic potential for heart regeneration, a rodent myocardial infarction model would seem more appropriate.

6. The ChIP data mapping SOX17 and EOMES binding to the promoter of PLGF is interesting, but given the focus of the manuscript on the importance of PLGF in promoting cardiovascular differentiation, arguably the downstream, rather than upstream mechanisms are the more relevant – does the hESC RNAseq data offer any insights on this?

Relatively minor concerns:

1. The manuscript is poorly written and requires thorough language correction. This may explain some of the ambiguities and why the key messages have not been communicated as clearly as the authors had intended.
2. The resolution of the figures provided is not adequate to permit a thorough review.
3. In the absence of suitable markers, it is unclear how the authors conclude localisation of PLGF to pericytes (Fig. 3e).

Responses to the Reviewers

We appreciate the time and consideration the *Nature Communications* journal reviewers took to review our manuscript in a stringent and fair manner. We are pleased to respond to their questions and concerns in a point-by-point fashion as below.

Reviewer #1

Comment #1

From the 52 growth factors, the authors selected 24 to generate modRNA from. From the manuscript, the rationale why these 24 is not clear and might need further explanation.

Response

Thank you for your suggestion. Based on the results in single-cell RNA-seq analyses of human and primate embryonic hearts, we selected the 24 growth factors to generate their modRNA as follows: 1) the top 12 growth factors (IGFBP5, VEGFA, FGF12, GDF10, SDF1 α , BMP4, IGF1, PLGF, TGFB2, IGF2, PDGFB, and TGFB1) that are highly co-expressed in the human ISL1⁺ SHF/OFT progenitor cells (Fig. 1e); 2) 6 growth factors (CLEC11A [SCGF], HGF, IGFBP4, NGF, NRG1, and PDGFA) that exhibit high association with CM and/or SMC/EC in Figure 1f; and 3) 6 growth factors (CSF3 [GCSF], FGF2, FGF4, FGF17, RLN2, and TNF α) that showed cardiogenic effects in the *in vitro* hESC-CM differentiation pilot studies, using each of recombinant proteins (data not shown). We described these points (page 7, line 151-157).

Comment #2

When PLGF modRNA was given 3 days after the initiation of CM differentiation, the number of TNNT2 expression cells increased. One explanation is indeed enhanced differentiation, another explanation can be increased proliferation so more cells can differentiate and/or differences in modRNA uptake as not all cells will be in the same differentiation and cell cycle state.

Response

Thank you for the input. When PLGF modRNA was given on day 3 in hESC-CM differentiation, the number of total cultured cells was not increased on day 6, compared to control. On flow cytometry analysis, we also found that the number and ratio of a cell-proliferation marker Ki67⁺ cells were not different between control and PLGF modRNA-treated cells on day 6. Irrespective of CMs (TNNT2⁺) or non-CMs (TNNT2⁻), the %Ki67⁺ ratios were unchanged in both groups. This is consistent with the results in Fig. 4c, in which PLGF deletion did not affect the %Ki67⁺ ratios in cultured cells in hESC-CM differentiation. Therefore, PLGF likely enhanced CM differentiation among the hESC-derived cells, rather than promoting CM proliferation. We described this point (page 8, line 162-165).

Comment #3

To determine the vasculogenic differentiation, the modRNAs were given to the cells 5 days after initiation of CM differentiation. Why day 5 and what is the percentage of EC/SMC differentiation in the day 6 treatment shown in fig 2c,d?

Response

We observed that the vasculogenic effects by modRNA emerged most significantly when modRNA was given on day 5 in hESC-CM differentiation, rather than on other days. This implies that the cardiac progenitor/intermediate cells cultured on day 6 may have already lost plasticity to differentiate into non-CM lineages, such as ECs, because when modRNA given on day 6 as in Fig. 2c,d, the ratios of an EC marker PECAM1⁺ and a SMC marker PDGFRB⁺ were <0.1-0.3% and <5-10%, respectively, which were lower than ones in the case where modRNA given on day 5.

Comment #4

Although very valuable and precious, it is not clear from figure 3e which cells do express PLGF. This reviewer is not convinced that ECs and pericytes are positive, it almost reflects a less mature CM.

Response

Thank you for the point. We repeated immunostaining of the human embryonic sectioned hearts. As shown in the new Fig. 3e and 3f, in the late-staged heart (≥ 8 weeks), PLGF expression was often seen in VE-cadherin (VEC)⁺ ECs/endocardial cells (Fig. 3e) and SM22⁺ SMCs/pericytes (Fig. 3f) in the heart, irrespective of their anatomical locations. These findings were contrasted with ones in the early-staged heart (Fig. 3d).

Comment #5

Comparing 3d and 3e, PLGF is very punctuated @w5.5, is it located in vesicles within the cell?

Response

As pointed, expression of PLGF, which was located in cytoplasm of the ISL1⁺ heart progenitors and TNNT2⁺ CMs in the early-staged heart (5.5 weeks), was somewhat spotty and punctuated (Fig. 3d). This is also confirmed in the newly added photos of the early-staged heart (5.5 weeks) in Fig. 3d, as shown below, and is contrasted with a bit denser expression in cytoplasm of the ECs/endocardial cells (Fig. 3e) and SMCs/pericytes (Fig. 3f) in the late-staged heart (≥ 8 weeks). This might be derived from the differential embryonic stages, amount of expression, and roles of PLGF in each of the cell types.

Comment #6

In figure 5 the authors show the RNA-seq analysis of hESC transfected with modRNA. Cell transfected with the LacZ RNA have a distinct profile from the WT cells at D4. This is a bit

worrisome. Would one not expect more overlap, and how does this influence the conclusions drawn?

Response

We understand the concern. As pointed, the LacZ modRNA-transfected cells at day 4 showed some variability on gene expression profile, although the LacZ modRNA cells at day 6 were quite similar to the WT cells at day 6. However, when comparing expression of the key genes on cardiomyogenesis and vasculogenesis, which were highlighted in the Extended Data Fig. 6, we found that the averaged values of these genes' expression were quite similar between the WT and LacZ modRNA cells at day 4, as shown in the following figure. In addition, direct comparison analyses with the limma package and the GSEA software were conducted using the RNA-seq data of WT, PLGF modRNA-transfected, and PLGF-KO cells at day 4 and 6 (Fig. 5c-j). Therefore, we think this influence would remain minimal.

Figure. Gene expression profiles of the 4 hESC-derived populations on day 4 in CM differentiation. Genes are the same as ones in the Extended Data Fig. 6.

Comment #7

Figure 6 nicely shows the *in vivo* differentiation potential of the cardiac progenitors with or without PLGF. Although a well conducted experiment and nice results, the expression and presence of PLGF is not confirmed. Is the ligand still present and which cell types express PLGF.

Response

Our previous studies revealed that modRNA-transfected cells could secrete the modRNA-derived protein until one week after transfection, at longest, both *in vitro* and *in vivo* (Rohner, et al. *Mol. Med.* 2021, 27: 102 / Yu, et al. *J. Control Release* 2019, 310: 103-114). In the kidney capsule experiments (Fig. 6), the engrafted kidneys were harvested at 1 month after injection of the hESC-derived heart progenitors with or without transfection of PLGF modRNA. Therefore, the exogenous expression of PLGF was likely diminished in the harvested kidney grafts. To confirm this, we performed immunostaining of the engrafted kidney that was injected by the PLGF modRNA-transfected heart progenitors. As shown below, expression of PLGF was not detected in the generated cardiac grafts.

Comment #8

Analysing the PLGF promoter, the authors identify several EOMES and SOX17 binding sites. They show that depending on the day post differentiation initiation, more SOX17 binding sites are used. Do these two transcription factor compete for promoter binding and what drives the binding of SOX17 to all 4 putative binding sites?

Response

The identified main binding site of EOMES on the PLGF promoter region (-1,934 bp upstream of the TSS) was different and far from those of SOX17 (Fig. 7a). In addition, the developmental stage when EOMES or SOX17 mainly binds to these identified sites on the PLGF promoter was different mutually (i.e., early or middle/late). Therefore, it appeared that EOMES and SOX17 would not compete for binding on the PLGF promoter, as we showed in Fig. 7d and described (page 16, line 369-371). Because SOX17 is known as a vasculogenic transcription factor (Kamachi & Kondoh, *Development*. 2013, 140:4129-4144) and was specifically expressed in the EC lineages of the primate embryonic hearts (Extended Data Fig. 7a-c), it is plausible that the upstream vasculogenic program would motivate SOX17 to bind onto the PLGF promoter, in order to promote EC differentiation and proliferation in a cell-autonomous fashion, as described in the response to the Comment #9.

Comment #9

Since PLGF signals via FLT-1/VEGFR1, is there co-expression or does PLGF influence intercellular differentiation?

Response

Thank you for the point. We investigated expression patterns of *PLGF* and *FLT1* (*VEGFR1*) in the single-cell RNA-seq datasets of the primate embryonic hearts and the *in vitro* hESC-derived cells during CM differentiation. As newly shown in Extended Data Fig. 7g-j, the *in vitro* early cardiac precursors (emerging on day 3 in hESC-CM differentiation; cluster #2 in Extended Data Fig. 7d) often co-expressed *PLGF* and *FLT1*, indicating the autocrine signaling and a cell-autonomous role of PLGF in these cells (Extended Data Fig. 7i,j). In the primate heart cells at 4 weeks of fetal age when the SHF cells (cluster #5 in Extended Data Fig. 7a) expressed *PLGF* specifically, while some of the SHF cells co-expressed *FLT1*, many

of the vascular progenitor cells (cluster #7 in Extended Data Fig. 7a) expressed *FLT1* but not *PLGF* (Extended Data Fig. 7g,h). This suggests that at the early stage *in vivo*, the SHF-derived PLGF would play a dual role, i.e., an autocrine role in the SHF and a paracrine role to affect the vascular progenitor cells for promoting intercellular vascular differentiation. In the primate heart cells at 7 weeks when the ECs and SMCs (cluster #11 and #13 in Extended Data Fig. 7a) expressed *PLGF* specifically, while ECs as well as endocardium (cluster #8) expressed *FLT1* strongly, SMCs showed little or no expression of *FLT1* (Extended Data Fig. 7g,h). Thus, it is suggested that at the late stage *in vivo*, the EC-derived PLGF would play a cell-autonomous (autocrine) role, whereas the SMC-derived PLGF would play a paracrine role to affect ECs and endocardium for promoting endothelial differentiation. We described these points (page 16-17, line 372-387).

Comment #10

Although not reported to be cardiomyogenic, PLGF has been reported to improve cardiac function. Has a potential cardiogenic effect been ignored in these studies?

Response

As pointed, PLGF has been reported to improve cardiac function in myocardial ischemia models (ref. #8,9,39) and a pressure-overload cardiac hypertrophy model (ref. #40) in rodents or pigs through several mechanisms, including enhancement of angiogenesis, reduction of oxidative stress and CM apoptosis, and/or functioning as a protective paracrine regulator to ECs, fibroblasts, and CMs. However, in these previous studies, murine or porcine adult hearts (but not neonatal hearts) were treated with PLGF, and it is well known there are few heart progenitor cells in the mammalian adult hearts (Bergmann, et al. *Cell* 2015, 161: 1566-75; Senyo, et al. *Nature* 2013, 493: 433-6). Therefore, one can expect that potential cardiogenic effects of PLGF would not appear clearly in those studies. In fact, our study showed that the PLGF's cardiogenic and cell-autonomous effects were clear only in early heart progenitors but not in differentiated CMs. We described these points (page 18, line 423-425).

Comment #11

In extended data figure 4, the authors treated PLGF-KO cells with PLGF to rescue the phenotype. Would one not expect that adding PLGF to the WT cells would improve CM differentiation as well?

Response

Thank you for the point. We repeated the *in vitro* hESC-CM differentiation experiments with treatment with 100 ng/mL of PLGF. As shown in the new Extended Data Fig. 5a and 5b, we found that treatment with 100 ng/mL of PLGF improved the CM differentiation efficacy in not only the PLGF-KO cells but also the WT cells.

Comment #12

In figure 1, the authors show the different populations present within the human and primate embryonic heart. When analysing the data shown in figure 1a,b, I do get the impression that the clusters are quite far separate from each other in 1a and from the genes shown in 1b, it is not very evident what cell population each cluster represent. In 1c, would one not expect that the Isl-1+ and LRG5+ cells to be more concentrated in one cluster (not necessarily the same)?

Response

Thank you for the point. In the human embryonic heart study (Extended Data Fig. 1), due to a

technical reason, e.g., manually picking of the dissociated single cells, the number of the sequenced and analyzed cells was relatively small. Therefore, some of the typical non-CM lineages, such as ECs and SMCs, needed to be excluded from further analyses because of their low abundance. This might cause some proximity among the analyzed cells and thus, potential limitation on interpretation of the obtained data in this study. We described this point in the Discussion section (page 20-21, line 475-479). On the other hand, the Guild-by-Association and correlation analysis (ref. #26) using the human embryonic heart single-cell RNA-seq data (Extended Data Fig. 1) revealed that *LGR5* expression was highly positively co-related to *ISL1* expression (corrected *P*-value: 5.86E-7) and therefore, *LGR5* was the 97th top-ranked gene associated with *ISL1*. In fact, although not all the cells, *LGR5* expression was relatively broadly overlapped with *ISL1* expression in the cluster #1 (CVP).

Comment #13

In figure 1g, another panel of genes is depicted to identify the clusters. This does make comparing the data in this plot with 1b complicated. Can the authors provide a more unified representation?

Response

We understand the point. However, due to the difference in the methods to harvest the dissociated single cardiac cells between the human (by manually picking) and primate (by sorting with FACS) studies, the numbers of the finally sequenced and analyzed cells were different in both studies. In addition, some of the typical non-CM lineages, such as ECs and SMCs, in the human heart studies needed to be excluded from further analyses because of their low abundance. Therefore, it would be theoretically somewhat difficult to combine and unify the two datasets. In the end, we decided to split the previous Fig. 1 and transfer the part of the human embryonic heart scRNA-seq analysis into the Extended Data Fig. 1, which is considered to corroborate the main findings in the primate heart scRNA-seq analysis in the new Fig. 1.

Comment #14

In figure 1k specific growth factors are identified per cell cluster. As these factors can only exert an effect if receptors are also present, can the authors indicate if they are present within the same cell cluster or in a neighbouring one.

Response

Thank you for the input and suggestion. This is a valid criticism. However, it would be complicated and difficult to show if the receptors are also present in the same cell populations for all of the growth factors and thus if the receptor-growth factor interactions exert biologically positive effects in the cells at each case, because some growth factors bind to not only one receptor but also two or more receptors with different affinities and sometimes distinct functions (e.g., BMPs – BMPR1A, BMPR1B, and BMPR2; FGFs – FGFR1, FGFR2, FGFR3, and FGFR4; TGFβs – TGFBR1, TGFBR2, and TGFBR3). In another case, some growth factor (e.g., FGF12) is not secreted but accumulated in the nucleus. Therefore, it would be too complicated to show the receptor-growth factor interactions' atlas for all of the around 50 growth factors. However, we newly showed this interaction map about PLGF with its specific sole receptor FLT1/VEGFR1 (Extended Data Fig. 7g-j), as described in the response to the Comment #9.

Reviewer #2

Comment #1

The scRNA-seq is based on 458 human cardiac cells, which divide into 10 distinct clusters, and 1,789 primate heart cells, with 12 clusters. The authors should indicate the number of cells in each cluster. The most relevant clusters appear to comprise fairly small numbers of cells. This might not be so concerning if not for the highly heterogeneous levels of expression within each cluster, as shown e.g. *Isl1*, *Hand1* on the feature plots in Fig.1.

Response

Thank you for the point. Following the suggestion, we showed the number of cells in each cluster in the human cardiac cells (Extended Data Fig. 1a) and the primate heart cells (Fig. 1a). As pointed, some clusters contained relatively small numbers of cells. Although this might be derived from the experimental size, due to a technical reason (e.g., manually picking of the dissociated human cardiac cells), and the rarity of those clusters (e.g., heart progenitors), it could be potential limitation on interpretation of the obtained data in this study. We described this point in the Discussion section (page 20-21, line 475-479).

Comment #2

It is unclear why the authors describe EC markers “PECAM1 and CDH5, clusters 8 and 11, to be endocardium and ECs, respectively. Extended Fig 1 shows there is little difference in expression of these genes in cluster 11, as expected. Surely, NPR3 should be highlighted as a marker of endocardium/ cluster 8? In this instance, I think it is an inaccurate description in the text, rather than inappropriate cluster annotation.

Response

Thank you for the input and suggestion. We completely agree that NPR3 should be highlighted as an endocardium marker for the cluster #8, while EC markers PECAM1 and CDH5 were enriched in both cluster #8 (endocardium) and #11 (EC), as expected. We modified the text to describe this point (page 6, line 129-132).

Comment #3

In contrast to point 2, I have queries over the cell type annotation of some clusters. For example, there is minimal overlap between NGFR and MYH11, both highlighted as markers of cluster 6 – this is not altogether surprising as neural crest cells would not be expected to express mature smooth muscle markers. As well as the low cell numbers and extent of transcriptional heterogeneity within clusters, the poor annotation of clusters is a concern. Given the prominence of the scRNAseq data in establishing the central hypothesis of the paper, these data need to be more accurately analysed.

Response

Thank you for the important point and suggestion. We re-analyzed the scRNA-seq data of the primate hearts and re-organized the cell type annotation of the segregated clusters (Fig. 1a,b and Extended Data Fig. 2). In fact, by elaborately modulating a resolution parameter for clustering of the analyzed single cells in the Seurat program, the old cluster 6 was segregated into the two clusters, the new clusters 6 and 13. The former was specifically enriched for expression of NGFR, while the latter was specifically enriched for expression of MYH11, suggesting that the new clusters 6 and 13 would represent neural crest cells and mature smooth muscle cells (SMCs), respectively (page 6, line 132-134).

Comment #4

The mechanism that the authors propose to be important is not explicitly explained. PLGF is expressed in cardiac progenitors as well as in cardiomyocyte, smooth muscle and endothelial derivatives. They detect a gain of function by ModRNA transfection and can rescue PLGF KO differentiation with recombinant PLGF protein. Thus, it is unclear whether the authors are proposing a paracrine role (which cell types secrete PLGF to induce progenitor differentiation?) or a cell-autonomous role – autocrine signalling?

Response

Thank you for the point. We investigated expression patterns of *PLGF* and *FLT1* (*VEGFR1*) in the single-cell RNA-seq datasets of the primate embryonic hearts and the *in vitro* hESC-derived cells during CM differentiation. As newly shown in Extended Data Fig. 7g-j, the *in vitro* early cardiac precursors (emerging on day 3 in hESC-CM differentiation; cluster #2 in Extended Data Fig. 7d) often co-expressed *PLGF* and *FLT1*, indicating the autocrine signaling and a cell-autonomous role of PLGF in these cells (Extended Data Fig. 7i,j). In the primate heart cells at 4 weeks of fetal age when the SHF cells (cluster #5 in Extended Data Fig. 7a) expressed *PLGF* specifically, while some of the SHF cells co-expressed *FLT1*, many of the vascular progenitor cells (cluster #7 in Extended Data Fig. 7a) expressed *FLT1* but not *PLGF* (Extended Data Fig. 7g,h). This suggests that at the early stage *in vivo*, the SHF-derived PLGF would play a dual role, i.e., an autocrine role in the SHF and a paracrine role to affect the vascular progenitor cells for promoting intercellular vascular differentiation. In the primate heart cells at 7 weeks when the ECs and SMCs (cluster #11 and #13 in Extended Data Fig. 7a) expressed *PLGF* specifically, while ECs as well as endocardium (cluster #8) expressed *FLT1* strongly, SMCs showed little or no expression of *FLT1* (Extended Data Fig. 7g,h). Thus, it is suggested that at the late stage *in vivo*, the EC-derived PLGF would play a cell-autonomous (autocrine) role, whereas the SMC-derived PLGF would play a paracrine role to affect ECs and endocardium for promoting endothelial differentiation. We described these points (page 16-17, line 372-387).

Comment #5

I do not understand the rationale for selecting a kidney capsule transplantation model to investigate *in vivo* differentiation of PLGF-treated progenitors. It seems an entirely artificial cellular environment in which to investigate cardiovascular differentiation. Moreover, no explanation is offered as to why early and late progenitors are combined in the assay. This may relate to the developmental mechanism that the authors envisage but it has not been explained. Given the discussion around therapeutic potential for heart regeneration, a rodent myocardial infarction model would seem more appropriate.

Response

We understand the point. However, a kidney capsule model is reported to be a good chimeric cell transplantation model to investigate *in vivo* cellular differentiation, as the model reflects general behaviors of the injected human cells well after transplanted into immunocompromised mice (D'Amour, et al. *Nat Biotechnol.* 2005, 23: 1534-1541). In fact, the histological appearances and degrees in differentiation and proliferation of the human heart progenitors after transplanted into both kidney capsules and hearts in immunocompromised mice were very similar and well correlated mutually between the two organs (Foo, et al. *Mol. Ther.* 2018, 26: 1644-1659 / Biermann, et al. *Stem Cells.* 2019, 37: 910-923). Another advantage in this model is to be less invasive, compared to a rodent myocardial infarction model, as shown in the differential survival ratios after surgery (100% vs 50-75% in our personal data). Thus, from the above-mentioned viewpoints and also the 3Rs (i.e.,

Replacement, Reduction, and Refinement)' principle in animal experiments, we chose a kidney capsule model to investigate the effects of PLGF modRNA-transfected heart progenitors for the *in vivo* chimeric heart muscle graft formation. We described this in the Discussion section (page 20, line 469-473).

Secondly, from a pilot study, we observed that injection of the combined early and late heart progenitors could generate bigger heart muscle grafts on kidney capsules than injection of only the single early or late heart progenitors. This is the reason why we combined early and late heart progenitors for transplantation into a kidney capsule. This may be derived from a better cell ratio of the CM-directed and non-CM (e.g., EC, SMC, etc.)-directed progenitor/intermediate cells in the combined approach. We described this point (page 13, line 303-305).

Comment #6

The ChIP data mapping SOX17 and EOMES binding to the promoter of PLGF is interesting, but given the focus of the manuscript on the importance of PLGF in promoting cardiovascular differentiation, arguably the downstream, rather than upstream mechanisms are the more relevant – does the hESC RNAseq data offer any insights on this?

Response

Thank you for the suggestion. With the hESC RNA-seq dataset (Fig. 5), we investigated the putative downstream mechanisms related to PLGF overexpression (with modified mRNA [modRNA]) or deletion during *in vitro* hESC-CM differentiation. As shown in the new Extended Data Fig. 6a, expression of a great number of the cardiomyogenesis drivers, such as typical transcription factors (e.g., HAND2, MEF2C, MYOCD, etc.) and chemical mediators (e.g., WNT, FGF, etc.), was significantly upregulated by treatment with PLGF modRNA while reversely downregulated by PLGF deletion, except for EOMES. This enhancement of the cardiomyogenesis programs was more predominant when the differentiating cells were treated with PLGF modRNA earlier (day 3 > 5). On the other hand, a number of vasculogenesis drivers (e.g., ETV1, APLNR, KDR, ANGPT1, etc.) were also significantly upregulated by treatment with PLGF modRNA while reversely downregulated by PLGF deletion, which was more predominant when the differentiating cells were treated with PLGF modRNA later (day 3 < 5), except for KDR (Extended Data Fig. 6b). These results suggest that PLGF would activate both the cardiomyogenesis and vasculogenesis programs directly and/or indirectly in a stage-dependent fashion. We described these points (page 12-13, line 280-290).

Comment #7

The manuscript is poorly written and requires thorough language correction. This may explain some of the ambiguities and why the key messages have not been communicated as clearly as the authors had intended.

Response

Thank you for the suggestion. We revised and brushed up the manuscript more carefully and entirely, so that the key messages could be clearer to the readers.

Comment #8

The resolution of the figures provided is not adequate to permit a thorough review.

Response

Thank you for the suggestion. We improved the resolution of all the figures in the manuscript, so that those could be clearer for reviewing.

Comment #9

In the absence of suitable markers, it is unclear how the authors conclude localisation of PLGF to pericytes (Fig. 3e).

Response

Thank you for the point. We repeated immunostaining of the human embryonic sectioned hearts. As shown in the new Fig. 3e and 3f, in the late-staged heart (≥ 8 weeks), PLGF expression was often seen in VE-cadherin (VEC)⁺ ECs/endocardial cells (Fig. 3e) and SM22⁺ SMCs/pericytes (Fig. 3f) in the heart, irrespective of their anatomical locations. These findings were contrasted with ones in the early-staged heart (Fig. 3d).

Finally, we again appreciate the reviewers' valuable comments and suggestions that encourage us to further improve the quality of the manuscript.

REVIEWERS' COMMENTS

Reviewer #1 (Remarks to the Author):

The authors, in this revised manuscript entitled Placental Growth Factor Exerts a Dual Function for Cardiomyogenesis and Vasculogenesis during Heart Development, addressed many of the concerns raised. There are some minor issues to address.

1. As most of the data is related/derived from stem cell differentiation as a model for heart development, it is advisable to adapt the title accordingly.
2. Page 4 line 85-88 refers to an assay developed to study the endogenous differentiation capacity of human embryonic cells since the derivation of chimeric embryo's as one would do with mouse EC cells is not allowed. As it is described in this manuscript, one might conclude human heart muscle transplantation, which is not the case but rather an in vivo differentiation assay. Please adapt accordingly throughout the manuscript.
3. page 5 line 100-101: the authors write in the rebuttal and manuscript that PLGF does not influence mature cardiomyocytes. It is therefore not clear how the authors envision PLGFmodRNA would be a promising therapy for heart disease.

Reviewer #2 (Remarks to the Author):

The authors have adequately addressed most of the comments raised by the reviewers.

My only remaining query is on the evidence for PLGF localisation to pericytes. The scRNAseq show convincing expression of PLGF in the smooth muscle clusters but it has not been demonstrated that these clusters also contain pericytes. In validation experiments, SM22a is used to infer expression in pericytes. However, SM22a has a broad expression pattern in early cardiomyocytes (Li et al, Circulation Research. 1996;78:188-195_, as well as smooth muscle cells and epicardium-derived cells following EMT.

What is the basis for concluding that pericytes are present in the scRNA-seq and that pericytes express PLGF? It does not detract from the manuscript to conclude that PLGF is predominantly expressed in SHF, CNCC, EC and SMC. Unless there is compelling evidence for pericyte expression, such statements should be removed.

Responses to the Reviewers

We appreciate the time and consideration the *Nature Communications* journal reviewers took to review our manuscript in a stringent and fair manner. We are pleased to respond again to their questions and concerns in a point-by-point fashion as below.

Reviewer #1

Comment #1

As most of the data is related/derived from stem cell differentiation as a model for heart development, it is advisable to adapt the title accordingly.

Response

Thank you for your suggestion. But we would like to remind the reviewer of the fact that all the single-cell RNA-seq data were derived from human and non-human primate developing hearts in this study. The histological data were also obtained using human embryonic hearts (Figure 3). Although other data were associated with and/or derived from *in vitro* hESC differentiation, this was also a model for heart development, as the reviewer pointed. We once thought if the title would be changed into “Placental growth factor exerts during cardiac differentiation and development”, but this was a bit too long and verbose. Finally, we decided that the original title should be kept.

Comment #2

Page 4 line 85-88 refers to an assay developed to study the endogenous differentiation capacity of human embryonic cells since the derivation of chimeric embryo's as one would do with mouse EC cells is not allowed. As it is described in this manuscript, one might conclude human heart muscle transplantation, which is not the case but rather an *in vivo* differentiation assay. Please adapt accordingly throughout the manuscript.

Response

Thank you for the input. As the reviewer pointed, the *in vivo* assay was not a model to transplant human heart muscles into mice but rather an assay investigating the *in vivo* differentiation capacity of human heart progenitor cells onto murine kidney capsules. Because the term, “human-mouse chimeric heart muscle grafts” was somewhat misleading, we revised that into “human heart progenitor-derived cardiac muscle grafts” throughout the manuscript.

Comment #3

page 5 line 100-101: the authors write in the rebuttal and manuscript that PLGF does not influence mature cardiomyocytes. It is therefore not clear how the authors envision PLGFmodRNA would be a promising therapy for heart disease.

Response

We would like to remind the reviewer that we do not write that PLGF does not influence mature cardiomyocytes. Rather, as shown in the *in vivo* differentiation assay with human heart progenitors (HPs) on murine kidney capsules (Figure 6e,f), we reported that areas that were positive for the cardiomyocyte maturation marker MLC2V were larger in the PLGF modRNA-transfected HP-engrafted group when compared to controls. These results indicate that PLGF modRNA promotes cardiac maturation in the *in vivo* developed heart muscle grafts.

Reviewer #2

Comment #1

My only remaining query is on the evidence for PLGF localisation to pericytes. The scRNAseq show convincing expression of PLGF in the smooth muscle clusters but it has not been demonstrated that these clusters also contain pericytes. In validation experiments, SM22a is used to infer expression in pericytes. However, SM22a has a broad expression pattern in early cardiomyocytes (Li et al, Circulation Research. 1996;78:188–195, as well as smooth muscle cells and epicardium-derived cells following EMT.

What is the basis for concluding that pericytes are present in the scRNA-seq and that pericytes express PLGF? It does not detract from the manuscript to conclude that PLGF is predominantly expressed in SHF, CNCC, EC and SMC. Unless there is compelling evidence for pericyte expression, such statements should be removed.

Response

Thank you for the input and suggestion. We agree the reviewer's point that SM22 is not necessarily a specific marker for pericytes, although frequently used as a SMC marker. Therefore, we removed the term "pericytes" on line 175 (page 8) and 973 (page 43).

Finally, we again appreciate the reviewers' valuable comments and suggestions that encourage us to further improve the quality of the manuscript.